

# Maximum ozone concentrations in the southwestern US and Texas: Implications of growing predominance of background contribution

David D. Parrish[1], Ian C. Faloona[2,3], and Richard G. Derwent[4]

5   1 David.D.Parrish, LLC, 4630 MacArthur Ln, Boulder, Colorado, USA

2 Air Quality Research Center, University of CA, Davis, CA, USA

3 Department of Land, Air, & Water Resources, University of CA, Davis, CA, USA

rdscientific, Newbury, Berkshire, UK

*Correspondence to*: David D. Parrish (david.d.parrish.llc@gmail.com)

**Abstract.** We utilize a simple, observational-based model to quantitatively estimate the US anthropogenic, background and wildfire contributions to the temporal and spatial distributions of maximum ozone concentrations throughout the southwestern US, including Texas and parts of California. The very different temporal variations of the separate contributions provide the basis for this analysis: over the past four decades the anthropogenic contribution has decreased at an approximately exponential rate by a factor of ~6.3, while the US background concentration rose significantly through the 1980s and 1990s, reached a

maximum in the mid-2000s, and has since slowly decreased. We primarily analyze ozone design values (ODVs), an extreme value statistic of relatively rare maximum ozone concentrations upon which the US National Ambient Air Quality Standards (NAAQS) are based; ODV time series provide spatially and temporally resolved records of maximum ozone concentrations throughout the country. Recent contributions of US background ozone to ODVs (primarily due to transported baseline ozone) are 64 to 70 ppb over most of the southwestern US, and wildfires (also generally considered a background contribution) add

further enhancements of 2 to 6 ppb in southwestern US urban areas. US anthropogenic emissions from urban and industrial sectors now produce only relatively modest enhancements to ODVs (less than ~6 ppb in 2020) outside of the three largest urban areas considered (Dallas, Houston and Los Angeles), where the 2020 enhancements were in the 17 to 30 ppb range. As a consequence, US background ozone concentrations now dominate over US anthropogenic contributions in the western US, including the Los Angeles urban basin, where the largest US ozone concentrations are observed. This finding has several

implications: 1) A pronounced shift in the spatial distribution of maximum US ozone concentrations has occurred; once ubiquitous nearly nationwide, ODVs of 75 ppb or greater have nearly disappeared in the eastern US, but are still frequent in the southwestern US. 2) By 2021, the trend of maximum ODVs in two major eastern urban areas (i.e., New York City and Atlanta) had decreased to the point that they were smaller than those in smaller southwestern US urban areas, and nearly as small as ODVs recorded at isolated rural southwestern US sites. 3) Together, the US background plus wildfire contributions

approach or exceed the US NAAQS for ozone of 70 ppb (implemented in 2015) and 75 ppb (implemented in 2008); consequently, in the southwestern US NAAQS achievement has been precluded. 4) Alternate emission control approaches





may provide more effective approaches to air quality improvement; since background ozone makes the dominant contribution to even the highest observed concentrations, an international effort to reduce northern midlatitude baseline ozone concentrations could be pursued, or a standard based on the anthropogenic increment above the regionally-varying US

background ozone concentration could be considered to provide a regionally uniform challenge of standard achievement. 5) The predominant contribution of US background ozone across the southwestern US presents a profound challenge for air quality modelling, since a manifold of stratospheric and tropospheric processes occurring on small spatial scales, but over hemisphere-wide distances, must be treated in detail to predict present and future background contributions to daily maximum ozone concentrations at local scales.

**1 Introduction and Background**

Elevated ambient ozone ($O_3$) concentrations constitute an air quality issue that has affected many urban areas of the world; in Los Angeles they reached extreme levels, with maximum 1-hr average urban ozone concentrations exceeding 500 ppb in the mid-1960s (Parrish and Stockwell, 2015). In the US, large reductions in anthropogenic emissions of photochemical ozone precursors, following implementation of air quality improvement policies, substantially lowered urban ozone concentrations

throughout the country over the past half century. However, several areas still have not attained the National Ambient Air Quality Standard (NAAQS) for ozone (see https://www.epa.gov/green-book). The NAAQS, most recently lowered in 2015, requires that the Ozone Design Value (ODV) at each monitoring site in a region not exceed 70 parts-per-billion (ppb, equal to nmole ozone/mole air). Notably the 2008 NAAQS is still in effect; even though it allows a higher ODV (75 ppb) there are different sets of nonattainment areas designated under the two standards. The ODV is an extreme value statistic defined as the

3-year average of the annual fourth-highest daily maximum 8-hour average (MDA8) ozone concentration. The fourth-highest MDA8 represents ~98th percentile of MDA8 values observed in the warm half of the year when those four highest values generally occur; thus, a time series of ODVs observed at a particular monitoring site is a smoothed record of the temporal evolution of the maximum ozone concentrations impacting that location. Ozone monitoring within the US began in the early 1970s, so ODVs have been collected over more than four decades at more than 2000 sites throughout the Nation. This

observational record reflects detailed, spatially and temporally resolved information regarding the variability of ozone sources and sinks that determine the maximum observed ozone concentrations. Our goal in this paper is to analyze this record to quantify the sources of maximum ozone concentrations in the southwestern US and Texas, and to investigate the implications of the results.

Our analysis utilizes a simple, observation-based quantification based on a conceptual model of the ozone sources that

determine ODVs throughout the US. This approach has been used previously (Parrish et al., 2017; 2022; Parrish and Ennis, 2019); it focuses on the two major contributors to ODVs in both urban and rural areas. The first is the ***US background ODV*** (i.e., the ODV that would be recorded at a site in the absence of any contribution from US anthropogenic emissions); this is the natural and anthropogenic ozone transported from outside the country, plus any ozone produced within the US from



naturally emitted precursors. The second is the ***US anthropogenic ODV enhancement*** (i.e., the enhancement of an actual
ODV above the US background ODV due to contributions from US anthropogenic emissions); this enhancement is due to
photochemical production from US anthropogenic emissions of ozone precursors. These definitions are consistent with the US
background ozone (USB) defined by the US Environmental Protection Agency (US EPA), a quantity designed to represent the
influence of all sources other than US anthropogenic emissions (e.g., Dolwick et al., 2015). We separately quantify the US
background ODV and the US anthropogenic ODV enhancement from tabulated ODV time series based on their very different
long-term temporal variability. Parrish et al. (2017) show that the US anthropogenic ODV enhancement has decreased rapidly
over the past four decades, a decrease accurately captured by an exponential term with an e-folding time of ~ 22 years; this
corresponds to a decrease of a factor of more than 6 from 1980 to 2020. In contrast, the US background ODV has remained
relatively constant. Parrish et al. (2017) and Parrish and Ennis (2019) assumed that it did remain constant, while Parrish et al.
(2022) used the measured temporal dependence of baseline ozone at northern midlatitudes (Parrish et al., 2020) to quantify the
relatively small changes that have occurred in the US background ODV near the US West Coast. The justification for this
quantification is that baseline ozone (i.e., ozone measured in air transported to a location without recent local or regional
anthropogenic or continental influences) represents the ozone concentration in air transported into North America from the
Pacific marine environment; it is this ozone that constitutes the large majority of the US background ODV. The text box below
(which is very similar to that included in Parrish et al., 2022) collects the important terms discussed above with their definitions;
parameters introduced later are also included.

---

**Important quantitative terms and parameters with definitions**

- **MDA8** - Maximum daily 8-hour average ozone concentration at a site on a particular day.
- **Ozone Design Value (ODV)** - 3-year running mean of the annual fourth-highest MDA8 ozone concentration at a site.
- **Baseline ozone** - The ozone concentration measured in air transported to a location without recent local or regional anthropogenic or continental influences.
- **US background ozone (USB)** - The ozone concentration that would exist at a time and US location from all sources, except that produced from US anthropogenic emissions.
- **US background ODV** – The ODV that would be recorded at a site in the absence of any contribution from US anthropogenic emissions.
- **US anthropogenic ozone (or ODV) enhancement** – The enhancement of an actual ozone concentration (or ODV) above the US background due to contributions from US anthropogenic emissions.
- *a* – Fit parameter of Equations 1, 3 and 4 that estimates the US background ODV in year 2000.
- *A* – Fit parameter of Equation 3 that estimates the US anthropogenic ODV in year 2000.
- *τ* – Fit parameter of equation 3 that estimates the exponential rate of decrease of the US anthropogenic ozone (or ODV) enhancement with time.

---

It is important to draw a careful distinction between the definitions of US background ODV and US anthropogenic ODV
enhancement presented above, and their observation-based quantifications that we conduct. Since those quantifications rely
on the observed decrease in ODVs, the term "anthropogenic" applies only to sources of precursors controlled by regulatory
action. The impact of uncontrolled anthropogenic precursor emissions, such as those from agricultural soils, livestock
operations, or volatile consumer products, for example, would be included as "background". Drawing conclusions from the



operationally derived quantifications must carefully consider this issue, and bring in additional considerations to avoid confounding influences. Furthermore, the ODVs represent rare events that may occur at any time during the warm season; thus, the two contributions are not necessarily exactly additive for any given extreme ozone episode; Section S2 of the Supplement discusses this issue in detail. Nevertheless the US background ODV (when accurately determined) does provide a direct quantification of the minimum ODV that could be achieved by reducing US anthropogenic precursor emissions alone, and the US anthropogenic ODV enhancement does provide a direct quantification of the enhancement of the actual ODV above that mimimum..

Parrish et al. (2022) applied the above referenced observation-based analysis to the entire western US coastal region; they quantify a small negative latitude gradient (-0.4 ppb/deg) and a strong positive vertical gradient (~10-15 ppb/km) in the lower troposphere along the US west coast that is weakened by continental mixing as marine air is advected eastward across the complex terrain of the North American Cordillera. The latitude gradient is consistent with earlier model and observational analysis (Zhang et al., 2020; Ziemke et al., 2011), and the altitude gradient is in excellent agreement with those determined by sondes and aircraft profiles along the California coast (Oltmans et al., 2008; Cooper et al., 2011; Yates et al., 2015; Faloona et al., 2020). The quantification of such fine scale variations (on the order of 1-2 ppb) provides a sense of the analysis precision, and the agreement with other analyses is evidence that the results of the observation-based quantification are physically realistic.

The ozone distribution in the troposphere results from the interactions of a great many chemical and physical processes driving ozone sources, sinks and transport. Fully understanding this distribution requires that the impacts of each of those processes be accurately simulated and results fully integrated with each other. Chemical transport models (CTMs) attempt this challenging task through computational parameterizations of each process in concert with the intention of reproducing broad observed ozone characteristics, yet model biases are quite often found to be in the range of 5-20 ppb (Hu et al., 2017; Zhang et al., 2020). In contrast, our observation-based approach avoids detailed process simulation by analyzing the measured ozone distribution, which necessarily reflects the accurately integrated impacts of all relevant processes, because the atmosphere itself has performed that integration. Comparison of results between different model approaches demonstrates that large uncertainties remain in CTM simulations (Section S1 of the Supplement briefly discusses model result comparisons). These comparisons have led to the increasing recognition that CTMs are not yet able to provide accurate estimates of atmospheric ozone concentrations without imposing additional constraints directly from observations. For example, Skipper et al. (2021) present a method to fuse ozone observations with US background ozone simulated by a regional CTM in order to correct model bias, and Hosseinpour et al. (2023) present a machine learning technique that improves CTM estimates of US background ODVs, which led to improved agreement with the Parrish et al. (2017) estimate for Los Angeles and the Parrish and Ennis (2019) estimate for New York City.

The results of the observation-based model applied in this work are also significantly uncertain, which arises from the underlying assumptions required to interpret observations so as to extract information regarding the processes that produced the observed ozone distribution. Notably, the simplicity of this model is helpful, because impacts of the model assumptions





can be readily evaluated; Sections S2-S6 of the Supplement present some of these evaluations. In contrast, CTMs require a manifold of parameterizations to simulate the many relevant chemical and physical processes; these parameterizations are generally embedded within the model computer code, rendering it extremely difficult to evaluate the uncertainty that arises from any specific parameterization. For example, evaluations by Derwent et al. (2016; 2018; 2021) identify important uncertainties in CTM results; however, determining avenues to correct the model deficiencies was not possible.

It is our experience that neither the observation-based model nor CTMs alone can provide an accurate and widely accepted quantification of the global ozone distribution. Given the uncertainty that arises in any one model, Derwent et al. (2023) argue that a hierarchy of models is required to provide robust, reliable support for ozone air quality policy development. In this regard, Held (2005) makes two key points:

• "It is fair to say that we typically gain some understanding of a complex system by relating its behavior to that of other, especially simpler, systems. For sufficiently complex systems, we need a model hierarchy on which to base our understanding, describing how the dynamics change as key sources of complexity are added or subtracted."

• "Elegance versus Elaboration. An elegant model is only as elaborate as it needs to be to capture the essence of a particular source of complexity, but is no more elaborate."

Emanuel (2020) makes similar points from a somewhat different perspective:

• "As it becomes easier to undertake complex computer simulations of climate and weather, … it is tempting to use computers to simulate, rather than understand, nature."

• "It is sometimes perceived as easier to run the model than to develop a comprehensive and satisfying understanding of the phenomena in question."

The observation-based model we use in this work is certainly on the "elegant" end of the required hierarchy, but it does indeed "capture the essence of a particular source of complexity"; in this work that complexity is the interaction of transported background ozone with anthropogenic enhancements to establish observed ozone concentrations. The simplicity of our model allows direct, conceptual understanding of the phenomena that underlie the observed ozone distribution, and this understanding provides a context for the beautifully detailed results provided by CTM simulations.

In this work we extend the observation-based analysis of ODV time series to the southwestern US, which we consider to be the Four Corner states of Utah, Arizona, Colorado and New Mexico, plus Nevada) and Texas, which is a region of particularly high US background ozone (Zhang et al., 2020); additionally, we also survey surrounding states, most importantly some of the California air basins. This quantification of the spatial and temporal variability of the US background ODV is important to improve our understanding of the magnitude and temporal changes of the primary ozone

sources, to quantify how the growing dominance of background ozone has changed the regional distribution of occurrence of ozone air quality standard exceedances, and to discuss the implications for future efforts to improve US ozone air quality. Notably, the US background ODV represents the minimum NAAQS that could possibly be achieved if the US anthropogenic ODV enhancement were reduced to zero by eliminating all US anthropogenic precursor emissions. Previous work shows that the US background ODV in some western US regions exceeds 60 ppb (e.g., Langford et al., 2022), and even has approached





70 ppb, making achievement of the 70 ppb NAAQS quite difficult in those regions (Cooper et al., 2015). In the following, Section 2 describes the data set that is the basis for this analysis, Section 3 discusses the analysis approach, Section 4 presents the results, and Section 5 discusses the conclusions and implications.

## 2 Data sets analyzed

This work examines two data sets: MDA8 ozone concentrations measured during the 1980-2022 period in four coastal
California air basins, and ODVs reported from the beginning of US ozone monitoring in the early 1970s through 2021 throughout the continental US. The California MDA8 ozone concentrations were obtained from California's Air Resource Board data archive (https://www.arb.ca.gov/adam/index.html). The ODVs are computed and published annually by US EPA's Office of Air Quality Planning and Standards. For each year for each ozone monitoring station in the US, an ODV is calculated if the measurements achieve the specified completeness criteria. All values reported in the focus region were downloaded from
the EPA data archive (https://www.epa.gov/aqs); only the ODVs marked as valid were retained for analysis. Exceptional events, such as wildfires or stratospheric ozone intrusions can, in principle, be removed from the MDA8 monitoring record, as uncontrollable "exceptional events", thereby altering the ODV archive; however, the analysis presented in this paper is not significantly affected. Section S6 of the Supplement more fully discusses this issue.

    Notably, very few sites report continuous MDA8 or ODV records over the nearly five decade period of ozone monitoring,
with some sites operating for only a few years. Generally, we include all reported values in our analysis, even if only a single ODV was reported from a particular site; any temporal discontinuity associated with initiation and termination of an individual site is assumed to still allow an accurate quantification of temporal trends within the selected region. Sensitivity tests have shown that analyses based on single sites with continuous records over the entire time period agree well with analyses of records combined from multiple sites with shorter duration records within the same region.

The ozone data considered here include the time period of emissions reductions resulting from societal efforts to control the COVID-19 pandemic. Many publications have examined the impacts of those emission reductions on ozone in areas throughout the world, with widely varying findings. No consistent impact has been found for summertime maximum ozone concentrations (e.g., Gkatzelis et al., 2021), which are the focus of this study. We find no apparent systematic deviations in MDA8 concentrations or ODVs reported for 2020 – the year of largest emission reductions – in any of the time series examined,
so this issue is not considered further in this analysis.

## 3 Methods

We analyze long-term changes in ODVs recorded during the entire periods of ozone monitoring in the southwestern US, surrounding states, and in two contrasting states in the eastern US. Since the ODVs are 3-year averages, the time series are insensitive to diurnal and seasonal variations, but do reflect changes on decadal (systematic long-term changes) and sub-





decadal (i.e., interannual variability) time scales. Fits of simple, continuous functional forms to the time series provide objective quantification of the average long-term changes. The parameter values derived from the fits provide that quantification, and are the basis for discussion of regional similarities and differences in ODV time series.

We also analyze the distribution of maximum MDA8 ozone concentrations recorded in four coastal CA air basins; these concentrations vary on daily and seasonal time scales, as well as the longer time scales that affect the ODVs. Since our focus

is on the maximum ozone concentrations, we only consider the single largest MDA8 ozone concentration observed at any of the sites in a given air basin on each day during the ozone season, which we take as May through September. Note that the site recording the largest MDA8 in each basin can vary from day to day. This selection gives 153 MDA8 values in each year in each basin. From these 153 values, we characterize each year's distribution by calculating several percentiles of the distribution: minimum, 10th, 25th, median, 75th, 90th, and maximum. The time series of each of these percentiles can then be fit

to the same continuous functional form as fit to ODV time series. Parrish et al. (2016) utilized a similar approach in their analysis of maximum MDA8 ozone concentrations recorded in the South Coast Air Basin (SoCAB), one of the coastal CA air basins considered in this work.

Quantified uncertainties of the parameter values derived from the functional fits are important in our analysis; 95% confidence limits are utilized, unless indicated otherwise. These confidence limits are derived from the fitting procedures

utilized in the analysis. The confidence limits from the ODV fits are widened to account for the known covariance between the recorded ODVs. Each ODV is a three-year running mean, so only every third ODV is independent from the others at a given site. Consequently, the number of independent ODVs in each fit is approximately a factor of three smaller than the number of reported ODVs, and the fitting routines therefore underestimate the true confidence limits of the derived parameters. All reported confidence limits are increased by a factor of $3^{1/2}$ to account for this covariance (Parrish et al., 2022). There may

exist additional sources of covariance (regionally coherent interannual variability, temporal interannual variability, etc.) between values included in any particular fit; we cannot account for the influence of any such additional covariance.

Networks of ozone monitoring sites have operated in the largest US urban areas; the ODVs from several of these networks are examined in detail. Generally, the full time periods of the data records are included in the analyses. However, in some data sets the early data do not appear to accurately represent the overall urban area; in this case the ODVs selected for analysis

begin at a later year. The most extreme example is the Las Vegas data set, where several newly established measurement sites began reporting ODVs in 2000; these values were significantly larger than reported from any other sites in that urban area. Thus, the analysis of the Las Vegas data is limited to 2000-2021 to avoid a discontinuity in the time series of ODVs.

## 3.1 Quantification of temporal changes in baseline ozone concentrations

Westerly winds that carry midlatitude Pacific background air are expected to provide the predominant source for background

ozone in the Western US. In accord with the approach of Parrish et al. (2022) we take the quantified temporal change in ozone at northern midlatitude baseline sites to quantify the temporal change in the US background ODV.





The temporal changes in baseline ozone have been derived from ozone measurement records collected at sites free of recent continental influence. We have no a priori knowledge of the functional form of baseline ozone changes, so a time series of baseline ozone concentrations is fit with the first few terms of a power series. No particular functional form for the time

evolution is assumed; instead, a quantitative description of the average continuous, long-term change is obtained for any series of observations (Parrish et al., 2019). In effect, the data time series itself dictates the functional form of the long-term change. In practice, the power series fit is obtained through a non-linear, least-squares regression fit of the baseline ozone data to a polynomial, with retention of only the statistically significant terms to ensure that the time series is not over fit. In the analyses discussed here only the first three polynomial terms are required,

$$\text{baseline O}_3 = a + b\text{t} + c\text{t}^2. \tag{1}$$

No statistically significant higher order terms (i.e., those with coefficient 95% confidence intervals not containing zero) have been found in any of the time series of baseline ozone data considered in this work.

The time origin is chosen as the year 2000 (i.e., t in Equation 1 equals the year - 2000) so that it falls within the time span of the data series, which allows precise determination of the coefficients in Equation 1. These coefficients have direct physical

interpretations:

- $a$, with units ppb $O_3$, is the intercept of the fitted curve in 2000; it quantifies the magnitude of the mean baseline ozone concentration in that year.
- $b$, with units ppb $O_3$ yr$^{-1}$, is the slope of the fitted curve in 2000; it provides the best estimate of the (continually varying) time rate of change of baseline ozone in that year.

- $c$, with units ppb $O_3$ yr$^{-2}$, quantifies the constant curvature of the fit; it is equal to one-half of the time rate of change of the slope of the fitted curve.

For the time series considered in this work, the fits of Equation 1 have statistically significant, negative values for $c$, with ozone concentrations increasing early in the data record, reaching a maximum, and then decreasing at later times. Equation 2 gives the year of the maximum of the fitted curve,

$$\text{year}_{\text{max}} = -b/2c + 2000. \tag{2}$$

Importantly, since a power series fit is not based on a physical model of the observed temporal changes, the fitted function cannot be reliably extrapolated to times outside the observation period; indeed, such extrapolations diverge to arbitrarily large positive or negative values at much earlier or later times.

Parrish et al. (2020) derived numerical values for the second and third coefficients in Equation 1 - $b = 0.20 \pm 0.06$ ppb yr$^{-1}$

and $c = -0.018 \pm 0.006$ ppb yr$^{-2}$ - that were common to baseline ozone time series measured throughout northern midlatitudes. Substitution of these values into Equation 2 indicates that baseline ozone reached a maximum in ~ 2006, following a decades long increase, and has since decreased. These same coefficient values provide excellent representation of 28 published trend analyses of baseline representative data sets collected throughout the western US (Parrish et al., 2021). We use these same coefficient values in this work.



## 3.2 Quantification of temporal changes of ODVs and MDA8 distribution percentiles

Long-term changes of ODVs in US urban areas generally differ markedly from those quantified in baseline ozone observations; urban ODVs have decreased rapidly over past decades, but generally appear to be approaching a non-zero background. Equation 3 defines a simple functional form that captures the physical picture of a US background ozone concentration that slowly varies as do baseline ozone concentrations, plus a rapidly decreasing US anthropogenic ozone enhancement;

$$O_3 = a + bt + ct^2 + A\exp(-t/\tau)$$
$$= a + 0.20*t - 0.018*t^2 + A*\exp(-t/\tau), \qquad (3)$$

where the values of the $b$ and $c$ parameters derived by Parrish et al. (2020) have been substituted into the second equation. Here $A$ is the year 2000 magnitude of an exponential term designed to quantify the rapidly decreasing US anthropogenic ozone enhancement, $\tau$ is the time constant for the exponential decrease, and t equals the year - 2000 as in Equations 1 and 2. The choice of the exponential function is more fully discussed in Sections S3 and S4 of the Supplement. Regression fits of Equation 3 to ozone concentration time series to derive quantitative values of the $a$ and $A$ parameters is the primary basis for analysis in this work.

An important issue in the present analysis is the derivation of the value of $\tau$. The time series of urban ODVs investigated here do not allow simultaneous derivation of precise values for all three ($a$, $A$ and $\tau$) parameters of Equation 3. This difficulty is surmounted by assuming that the value of $\tau = 21.8 \pm 0.8$ years derived in earlier work for southern California (Parrish et al., 2022) is also appropriate for all regions considered here. A justification for this assumption is that precursor emission control strategies, particularly for on-road vehicles, implemented throughout the US have been similar to those in California; it is on-road vehicle emissions that have dominated urban ozone production over the past several decades (Nopmongcol et al., 2017). Parrish and Ennis (2019) also made this assumption for the northeastern US, and presented several consistency checks that show this assumption is appropriate for that region, which is further removed from southern California than the southwestern US considered here. Section S5 of the Supplement further discusses the justification for this assumption in the southwestern US and the uncertainty of the derived value of $\tau$.

## 3.3 Long-term temporal changes in wildfire contributions

ODV contributions from wildfire emissions can affect the results of analysis based on Equation 3. These emissions have not been systematically reduced - rather the wildfire emissions are believed to be increasing as the climate warms (e.g., Westerling et al. 2006; Westerling, 2016). Parrish et al. (2022) discuss three characteristics of the impact of wildfire emissions on ODV time series: first, wildfire emissions alone do not necessarily significantly elevate ODVs, but when they mix with local $NO_X$ emissions, such as in urban areas, their impact can more easily be discerned; second, wildfire impacts have systematically increased over past decades; and third, wildfires are highly sporadic, both temporally and spatially (Iglesias et al., 2022). Addition of another term to Equation 3 can approximately account for the systematic, temporally increasing influence of wildfire emissions. Burke et al. (2021) estimate that over the past four decades, the wildfire burned area in the US has roughly



quadrupled in an approximately linear fashion (their Fig. 1A). Similarly, Iglesias et al. (2022) estimate annual mean area burned in the western US has risen by 220%–330% across a 20 year span within the period from 1984-2018 (their Fig. 7A). We represent the ODV contribution from wildfires by a similar increase; a final, additional term in Equation 4 increases linearly

by a factor of 4 from 1980 to 2020, where the parameter $WF$ represents the location specific ODV enhancement due to wildfires in the year 2000.

$$ODV = a + 0.20*t - 0.018*t^2 + A_{WF}*exp(-t/\tau) + WF*(1 + 0.03*t) \qquad (4)$$

Here $A_{WF}$ represents a revised $A$ parameter in locations where enhancements of ODVs due to wildfire emissions are significant. The linear increase of the wildfire contribution does not account for the sporadic character of wildfires; however this functional

form is appropriate for the 3-year averaging period inherent in ODV time series. There are available data bases giving the spatial and temporal distributions of area burned by wildfires with detail that varies from annual means for the entire country (e.g., https://www.nifc.gov/fire-information/statistics/wildfires) to monthly means for individual states (e.g., monthly area burned in California from 1972 to the present is available from California Department of Forestry and Fire Protection). However, to incorporate such detailed information into our analysis would require knowledge of the spatial and temporal

variability of wildfire impacts at individual sampling sites, which is not available without detailed atmospheric transport modelling. Therefore, year-to-year variation of wildfire impacts undoubtedly contributes to the variability of ODVs about the functional fits to the ODV time series discussed in this work, and higher values of RMSD in any basin may be indicative of the prevalence of wildfire influence.

Fitting Equation 4 to an ODV time series generally does not allow precise determination of values for all three parameters

($a$, $A_{WF}$ and $WF$, with $\tau$ already fixed); however if $a$ can be determined from a separate analysis, then precise values can be determined for $A_{WF}$ and $WF$. We follow this approach in the examination of time series of maximum ODVs recorded in southwestern US urban areas. Additional complications can arise from ozone contributions produced from agricultural emissions or from other anthropogenic emissions that have not been as effectively controlled as other anthropogenic emissions. Parrish et al. (2017, 2022) discuss such ODV contributions in intense agricultural regions of California; using GEOS-Chem

Geddes et al. (2022) modeled significant $NO_X$ emissions from agricultural soils throughout the southwest but especially in Texas and California. Parrish and Ennis (2019) discuss possible ODV contributions from volatile chemical products; their emissions history is not well quantified, but are likely increasing (McDonald et al., 2018). Other ODV contributions that have not been controlled include those from oil and gas extraction processes, which are important in some regions of the southwestern US and have increased overall during the past decades. Depending upon their specific temporal dependence,

ODV contributions from these uncontrolled emission sources could contribute to the "wildfire" term in Equation 4 if they have been increasing, but none are expected to be doubling in magnitude every 20 years as the parameterized wildfire source is here. Section S4 of the Supplement further discusses some issues regarding uncontrolled anthropogenic ozone precursor emissions.





### 3.4 Additional analysis considerations

The ODV time series fit to Equations 3 and 4 are either from single sites or multiple sites that are in reasonably close spatial proximity and exhibit similar temporal changes. There is some subjectivity in this selection of time series, which is guided by maximizing the temporal length of the series, minimizing confidence limits of the derived parameters, and minimizing deviations between the ODVs and the fits.

      The quality of the fits of Equations 3 and 4 are quantified by root-mean-square deviations (RMSDs) between the fits and

the recorded data, which scatter both above and below the fits; these RMSDs are generally in the range of 3 to 5 ppb (e.g., Tables S1 and S3-S5). These deviations are attributed to quasi-chaotic, varying ODV contributions from wildfires, stratospheric intrusions, variable meteorological conditions, etc. Our observation-based model does not account for this residual variance.

      The exponential function is chosen to be consistent with our general understanding of US ozone changes (see discussion

in Section S3 of the Supplement). The parameter values can be directly related to the processes controlling US ozone concentrations; i.e., Equations 3 and 4 are designed to quantify the magnitudes of the US background ODV and the US anthropogenic ODV enhancement in the year 2000 by the values of the $a$ and $A$ (or $A_{WF}$), respectively. However, it must be recognized that this design does not necessarily guarantee that a parameter value can be directly interpreted as Equations 3 and 4 imply; the interpretation of the parameter values must proceed with careful consideration of potentially complicating factors,

as fully discussed in the sections below and the Supplement.

### 4 Results

We analyze the 4-plus decades long time series of ODVs recorded throughout the southwestern US and Texas and as well as in surrounding and more distant states, with the goal of estimating the spatial and temporal distributions of the two major contributions - US background ozone and the US anthropogenic ozone enhancement - to these observed concentrations. as

This analysis is based on Equations 3 and 4. We also analyze the full distribution of maximum MDA8 ozone concentrations recorded in four CA coastal air basins.

      Parrish et al. (2020) quantified the long-term baseline ozone change that has occurred throughout northern midlatitudes by determining the values of the $b$ and $c$ coefficients of Equation 1 through a fit of that equation to eight baseline ozone data sets collected at surface sites and from sondes and aircraft over western Europe and western North America. Here we use those

same coefficient values in Equation 3 to examine the ODV time series recorded in the southwestern US region. Parrish et al. (2022) applied this approach to ODV time series recorded in the western US coastal states; one source of uncertainty in that analysis is the question of whether the results of Parrish (2020), based upon mean baseline ozone measurements, can be appropriately applied to the maximum ozone measurements that define ODVs. Section 4.1 investigates the appropriateness of this assumption. Section 4.2 analyzes the temporal evolution of the distribution of the maximum MDA8 ozone concentrations





in the California coastal air basins, and the following three sections present the analysis of ODV time series in the southwestern US and Texas urban and rural areas, as well as some in other US states to provide context.

**4.1 Long-term ODV changes at isolated rural sites in the western US**

Investigation of time series of ODVs at remote rural sites provides an opportunity to quantify the long-term changes of ODVs in regions where the US anthropogenic ODV enhancements are small; in the limit of zero US anthropogenic ODV

enhancement, Equation 3 is reduced to Equation 1. Figure 1 shows ODV time series recorded at eight such sites included in the Clean Air Status and Trends Network (CASTNET) of the US EPA (https://www.epa.gov/castnet) whose locations are shown in inset map. They are chosen to span a wide latitude range (32° to 48.5° N), and to be as isolated as possible from anthropogenic emission sources. Seven are in a relatively narrow longitude band (109.4° to 114.2° W) located ~750 to 1100 km east of the US west coast, and the other is nearer (~220 km) the coast (Lassen Volcanic NP at 121.6° W). All are at similar

elevations (2.0 ± 0.43 km) except Glacier NP at 0.96 km.

Data from most of these same CASTNET sites have been included in previous studies of long-term ozone changes at western US rural and remote sites. Lassen Volcanic NP was investigated by Jaffe et al. (2003), Jaffe and Ray (2007), Parrish et al. (2009; 2012; 2017; 2020) and Cooper et al. (2014), due to the site's location near the US West Coast. Another four of the sites (Glacier NP, Yellowstone NP, Craters of the Moon NM and Canyonlands NP) were also considered by Jaffe and Ray

(2007). Two of the remaining sites (Great Basin NP and Grand Canyon NP) were included by Cooper et al. (2020) in their analysis of surface ozone trends at globally distributed remote sites. Here we also consider Chiricahua NM, a more southerly site in Arizona. To our knowledge, these data sets comprise all of the longest, high quality, relatively isolated rural data sets available in the western continental US.

The CASTNET time series of ODVs are illustrated in the upper graph of Figure 1. The curves are regression fits of Equation

1, which quantify the long-term changes of baseline ozone. Table S1 gives the derived parameter values, along with confidence limits and the root-mean-square deviations (RMSD) of the ODVs from the fits. The derived $a$ parameter values show that the absolute ODV values exhibit a systematic spatial gradient; there is close agreement among the largest $a$ parameter values at the five southern sites (weighted mean = 71.5 ± 0.8 ppb, standard deviation = 1.1 ppb), with values decreasing with latitude and/or elevation at the three more northern sites, reaching a minimum of 54.9 ± 1.1 ppb at Glacier NP, the most northern and

lowest elevation site. In contrast, the derived $b$ and $c$ coefficients at all sites agree within their confidence limits, which indicates the absence of a statistically significant difference in the temporal evolution of the ODVs among these widely separated sites, although substantial uncertainty remains in the determination of the coefficient values.

The similarity of the temporal evolution of these rural ODVs suggests normalizing the ODVs to remove the spatial gradient and to derive a single fit of Equation 1 to all ODVs from the eight sites. We choose to normalize to 71.4 ppb in the year 2000;

this normalization factor is the average of the $a$ parameters derived at Great Basin NP, Canyonlands NP and Grand Canyon NP (solid symbols in Figure 1), which are chosen to represent the central southwestern US. The 212 normalized ODVs recorded over 32 years at the eight sites are fit within a RMSD of 2.4 ppb (lower panel of Figure 1). This fit gives $b = 0.07 ± 0.13$ ppb





yr$^{-1}$ and $c$ = -0.015 ± 0.005 ppb yr$^{-2}$; Table S1 indicates that all of the $b$ and $c$ parameters derived from the separate site fits agree with these values within their confidence limits. These parameter values indicate that the rural ODVs increased early in the data record, reached a maximum in year$_{max}$ = 2002 ± 4, and decreased thereafter. A linear fit to all of the normalized ODVs over the 2000-2021 period indicates a statistically significant average negative trend equal to -0.30 ± 0.10 ppb/yr.

The $b$ and $c$ parameter values derived here from the normalized CASTNET ODVs agree (within derived confidence limits, see Table 1) with the parameter values derived for the entire northern midlatitudes (Parrish et al., 2020) from mean baseline ozone data collected at locations including marine boundary layer (MBL) and mountain surface sites as well as free troposphere data from sonde and aircraft. This agreement indicates that ODVs recorded at the eight CASTNET sites provide approximate, direct measures of southwestern US background ODVs and strongly support applying Equation 3 for analysis of ODV time series. The fit of Equation 3, which includes the exponential term, to the normalized CASTNET ODVs is indistinguishable from the fit of Equation 1 (black curve in the lower graph in Figure 1), although it does give a smaller value for the year 2000 US background ODV ($a$ = 68.5 ± 1.5 ppb, compared to 71.3 ± 0.8 for the fit to Equation 1) with a small, but significant anthropogenic ODV enhancement ($A$ = 2.8 ± 1.9 ppb). We interpret the final term of Equation 3 as the mean ODV enhancement due to transport of ozone from US anthropogenic precursors to these isolated rural sites in the western US.

In summary, the 2$^{nd}$ and 3$^{rd}$ terms of Equation 3 provide an excellent estimate for the long-term change of the mean US background ODV in the western US; these terms added to the derived $a$ parameter value of 68.5 ± 1.5 ppb, is plotted as the grey curve in the lower graph in Figure 1. Similar curves, normalized to $a$ parameter values derived in other fits, will be included in later graphs to illustrate the purely baseline contributions to ODV time series considered in the following analyses.

**4.2 Contributions to maximum MDA8 ozone concentration distributions in four coastal California air basins**

Previous analyses have shown that Equation 3 (or a closely related equation with a constant background term, rather than the small changes quantified by the 2$^{nd}$ and 3$^{rd}$ terms of Equation 3) gives excellent fits to ODV time series, not only in urban areas (Parrish et al., 2017, 2022, Parrish and Ennis, 2019), but also in relatively isolated northern rural states (Parrish et al., 2022) and at western US rural CASTNET sites (preceding section). Here we examine how well fit are the entire distributions of maximum MDA8 ozone concentrations in four coastal CA air basins. These basins are selected to span a wide range of environments: the intensely photochemically active, highly urbanized South Coast Air Basin (SoCAB, containing the Los Angeles urban area) and San Diego AB, the highly urbanized but less photochemically active San Francisco Bay AB (SFB AB), and the rural North Coast AB. Their locations are indicated in a map inset in Figure S1. Coastal air basins are selected to minimize impacts from uncontrolled anthropogenic emission sources, such as agricultural activity.

To quantify the overall distribution of the maximum ozone concentrations recorded each year in each air basin we examine the maximum MDA8 ozone concentration recorded at any site within the basin on each day in the May to September ozone season; this gives a total of 153 values each year in each air basin. We take these values as representative of the distribution of the basin maximum ozone concentrations that occur under all meteorological conditions of the ozone season. We quantify the year's MDA8 distribution in a basin by determining seven percentiles of those 153 values: minimum, 10$^{th}$, 25$^{th}$, median, 75$^{th}$,



90$^{th}$, and maximum. Figure 2 illustrates the temporal evolution of these percentiles; fits of Equation 3 to the time series of each percentile are included, with the derived parameter values given in Table S2. The fits are limited to the 36 year 1980-2015 period for most of the percentiles in the three urbanized air basins; later years are not included due to clear positive deviations of the higher percentiles of the distributions, most clearly in the SoCAB, whose cause(s) have not been established. One likely

caused is the rise of a new regime of wildfire impacts. According to CalFire (FRAP, https://www.fire.ca.gov/what-we-do/fire-resource-assessment-program) 2007 was the first time in recorded history that California lost over one million acres in a year to wildfires, but since then the four highest wildfire years have occurred in 2020, 2021, 2018, and 2017 respectively burning 4.1, 2.2, 1.8, and 1.3 million acres. Although exact matches cannot be expected without detailed analysis of where each wildfire had its impacts, these years, particularly 2020, generally stand out in the recent maximum values evident in Figure 2.  In

contrast, Kim et al. (2022) present a detailed investigation of these deviations in the SoCAB, and conclude that they are caused by substantial recent changes in the photochemical regime in the SoCAB. Note that the fits to the lowest percentile (i.e., the minimum MDA8) in the three urbanized air basins and to all percentiles in the rural North Coast Air Basin the fits in Figure 2 include the entire 43 year 1980-2022 period.

Despite the marked differences in the long-term changes between percentiles and basins evident in Figure 2, the fits of

Equation 3 provide excellent descriptions of the long-term changes for all percentiles in all four basins; all RMSDs of the data about the fits are in the 2.5 to 9 ppb range (Table S2), except for the maxima, which exhibit larger variability. As expected, the anthropogenic contributions (quantified by the derived $A$ parameter values) are largest in the SoCAB for all percentiles, except the minimum. The corresponding $A$ parameter values are approximately a factor of ~2 smaller in the San Diego AB than those in the SoCAB, and they are lower by another factor of ~2 in the SFB AB. In the North Coast AB all derived $A$

parameter values are near zero (-2.8 to 4.6 ppb); a significant negative $A$ parameter value indicates that reaction of transported background ozone with fresh NOx emissions has lowered observed MDA8 concentrations below those that would result from US background ozone alone. Importantly, the fits to the North Coast AB percentiles are all qualitatively similar to the fits to the CASTNET ODVs in Figure 1, exhibiting characteristic baseline ozone temporal behavior; the fits to the minima in all four ABs also show qualitatively similar behavior with small $A$ parameter values (<~ 5 ppb). This similarity indicates that

transported baseline ozone concentrations dominate the lowest percentiles of observed maximum MDA8 concentrations in all air basins, even in the SoCAB.

The derived $a$ and $A$ parameter values from the fits in Figure 2 define the distributions of US background ozone concentrations and US anthropogenic ozone contributions in the year 2000; the upper panel of Figure S1 are bar graphs illustrating those distributions. By 2015 the distributions of the two ozone contributions had evolved as described by Equation

3 to those shown in the lower graphs of Figure S1. The US anthropogenic contribution had decreased by a factor of 2.0 (= exp (15/21.8), based on the exponential term in Equation 3), while the US background ozone concentration had decreased by only 1 ppb (difference in the first three terms of Equation 3 between 2000 and 2015). Figure 3 compares those 2015 US anthropogenic contributions with the mean US background ozone concentration, which is derived from five separate US background ozone distribution estimates, as discussed further in Section 5.1.





### 4.3 ODV contributions in southwestern US

Four urban areas in the southwestern US – Phoenix AZ, Denver CO, Las Vegas NV and Salt Lake City UT– are centers of Marginal or Moderate Ozone Nonattainment Areas (see US EPA Green Book 8-Hour Ozone (2015) Area Information, last accessed 27 January 2023); additionally Phoenix AZ and Denver CO are classified as Moderate and Severe-15, respectively, Ozone Nonattainment Areas under the 2008 ozone NAAQS (see US EPA Green Book 8-Hour Ozone (2008) Area Information, last accessed 27 January 2023). We examine the ODV time series recorded in these four urban areas plus three smaller cities –Tucson AZ, Reno NV and the Albuquerque/Santa Fe NM area. The map in Figure 4 shows the ozone monitoring site locations in these seven urban areas. Monitoring sites also sparsely cover the rural areas in these five states (map in Figure S2); we also examine the ODVs recorded at these rural sites.

Figure 5 displays the analysis of the Denver CO urban (left graph) and Colorado rural (right graph) ODV time series. The urban ODVs generally fall above the fit to baseline data (black curve with dashed extension, which is reproduced from the grey curve in Figure 1). One exception is the Camp site with some ODVs falling well below the baseline curve; since this site is located at street level in central urban Denver, we attribute these small ODVs to reductions in observed ozone concentrations due to its reaction with fresh $NO_X$ emissions. The rural data generally scatter about the baseline curve. Two rural sites that have received attention in previous work are emphasized as solid symbols. Gothic CO in west central CO (location indicated in Figure S2), has been considered to be a remote site (Cooper et al., 2020); that ODV time series agrees with the baseline curve within a RMSD of 1.8 ppb. Rocky Mountain NP (location also indicated in Figure S2), has been considered to be a high-elevation baseline site (Lin et al., 2017). The green dashed curves show fits of Equation 3 to all ODVs in the urban (excluding the Camp site) and rural (excluding the Rocky Mountain NP site) data sets. The red curve shows the fit of Equation 4, which includes possible wildfire influences, to the maximum urban ODVs. The parameter values from these fits are annotated in the figure, and given to greater precision in Tables S3 and S4.

The derived $a$ parameter values from analysis of Denver urban ($69.0 \pm 2.1$ ppb) and Colorado rural ($69.0 \pm 3.1$ ppb) ODV are consistent with each other and agree closely with that from the fit to the normalized CASTNET data ($68.5 \pm 1.5$ ppb). The one exception is the Rocky Mountain NP site; the ODV temporal changes at this site are consistent with those of CASTNET data, but give a larger $a$ parameter value ($73 \pm 5$ ppb), which is possibly due to the relatively higher elevation of the Rocky Mountain NP site (2.7 km elevation) relative to the lower elevation of the Denver urban area (~1.8 km elevation), and/or to the influence of Denver area urban pollution, which has been demonstrated to impact this site (Evans and Helmig, 2017). The derived $A$ parameter values provide estimates of the US anthropogenic ODV enhancement in the year 2000 averaged over the sites in each group: $8.0 \pm 1.7$ ppb and $-1.9 \pm 4.4$ in CO urban and rural areas, respectively; note that this rural value is consistent with zero within the derived confidence limit. The fit of Equation 4 to the time series of maximum ODVs (with the $a$ parameter value fixed at 69.0 ppb) gives a somewhat larger $A$ parameter value ($A_{WF} = 11.0 \pm 1.7$) and a $WF$ parameter value of $4.0 \pm 2.5$ ppb; this latter value is an estimate of the ODV enhancement due to wildfire emissions in the year 2000.





Figure 6 shows the results of similar analyses for six other southwestern US urban areas. In these plots (and in Figure S3), the black, generally lower curves with dashed extensions are the fit to the baseline data from Figure 1, normalized to the respective $a$ parameter values; thus, these curves indicate the temporal behavior of ODVs in each data set that would have been expected in the absence of US anthropogenic emissions. The derived $a$ parameter values are similar in 5 of those areas, all falling in the range of 66.2 to 69.0 ppb, and agree well with the Denver ($69.0 \pm 2.1$ ppb) and CASTNET ($68.5 \pm 1.5$ ppb) results discussed above. Only in Tucson is a significantly smaller $a$ parameter value ($63.9 \pm 1.4$ ppb) derived; this value is also smaller than found at the nearby Chiricahua NM CASTNET site ($70.1 \pm 1.7$ ppb); the lower elevation of Tucson may contribute to this difference. Fits of equation 3 to all of the rural ODV time series (Figure S3 and Table S3) also give similar $a$ parameter values (within 65 to 69 ppb), with small $A$ parameter values (-3 to +5 ppb). Fits of equation 4 to the maximum urban ODVs (with the $a$ parameter values fixed at the values derived from the ODVs in the entire respective urban area) give $A_{WF}$ parameter values in the range of 6 to 16 ppb. Small, positive wildfire ODV enhancements ($WF$ = range 0.6 to 1.6 ppb) are derived in all six urban areas, although the confidence limits of all include zero.

In summary, analyses of the ODV time series throughout the southwestern US give remarkably consistent results. The weighted average $a$ parameter value, which gives an estimate of the US background ODV in the year 2000, derived from 11 urban and rural ODV time series, is $67.4 \pm 0.7$ ppb with only the low elevation, most southerly Tucson urban area giving a significantly lower value; this average agrees well with the value derived in the CASTNET site analysis ($68.5 \pm 1.5$ ppb). This overall agreement indicates that an approximately constant, common US background ODV value is found throughout the southwestern US. The mean $A$ parameter value, which gives an estimate of the US anthropogenic ODV enhancement, derived from four rural ODV time series, is $-0.1 \pm 2.9$ ppb, which indicates that ODVs outside of urban areas are little affected by US anthropogenic emissions throughout this entire region. Any area with intense agricultural activity may be an exception to this general finding, due to the influence of uncontrolled anthropogenic emissions from this activity. Within the seven urban areas, the $A_{WF}$ values span the range from 6 to 16 ppb, with the larger values (11 to 16 ppb) derived in the four nonattainment areas. Wildfire emissions contribute additional ODV enhancements within urban areas: estimated as 0.6 to 4.0 ppb (mean of $1.6 \pm 0.9$ ppb) in the year 2000, and 1 to 6.4 ppb (mean of $2.6 \pm 1.4$ ppb) in 2020.

### 4.4 ODV contributions in Texas

The state of Texas has four urban areas – Dallas, Houston, El Paso, and San Antonio – that are centers of marginal to moderate ozone non-attainment areas designated under the 2015 NAAQS (US EPA Green Book 8-Hour Ozone (2015) Area Information, last accessed 31 January 2023); additionally Houston and Dallas are classified as Severe-15 non-attainment areas under the 2008 ozone NAAQS (US EPA Green Book 8-Hour Ozone (2008) Area Information, last accessed 31 January 2023). We examine the ODV time series recorded in these four urban areas, plus five other Texas regions containing smaller cities as well as vast rural areas. Figure 7 indicates the ozone monitoring site locations throughout the state, plus some sites in neighboring states, and identifies the division of these sites among nine Texas regions and the state of Oklahoma; we examine the ODVs recorded in these regions. Figure 8 plots the ODV time series recorded in seven of the Texas regions, and Figures




S4 and S5 give similar plots for the two remaining Texas regions, for Oklahoma, and for eight additional surrounding states. The parameter values derived from fits of Equations 3 and 4 to all ODV time series are annotated in the figures, and given to greater precision in Tables S4 and S5. In Figure 8 and in the upper graphs of Figure S4 the ODVs for the entire state of Texas (grey symbols) are included in the plots highlighting the regional ODVs (colored symbols), which are fit to Equation 3 as indicated by the upper black solid curves. Lower black curves with dashed extensions indicate the fit to the baseline data from Figure 1, normalized to the respective $a$ parameter values; these curves indicate the temporal behavior of ODVs in each region that would have been expected in the absence of US anthropogenic emissions.

Comparison between Figures 6 and 8 (note differences in ordinate scale) indicates that ODVs in Texas were often much larger in earlier decades than those in the southwestern US urban areas. These larger ODVs indicate much larger impacts from US anthropogenic emissions in Texas than in the other southwestern US states. For example, in 2000 the maximum US anthropogenic ODV enhancements (i.e., $A_{WF}$ parameter values) in Texan urban areas were generally > 25 ppb, and as much as 54 ± 3 ppb in the Houston region, compared to the in the range of 6 to 16 ppb found in the Southwestern US. However, the ODVs in Texas have decreased by a larger absolute amount than in the southwestern US, so that at present the maximum ODVs in Texas are no larger than those recorded in the Denver and Phoenix urban areas.

The southwestern US is characterized by an approximately constant US background ODV across the entire region, but across Texas there is a pronounced spatial gradient. The $a$ parameter values of 65 ppb derived for the El Paso and Western Rural regions are similar to the weighted mean (67.4 ± 0.7 ppb) of all of the urban and rural southwestern $a$ parameter values (excluding the fits to the urban maxima) in Table S3. However, the US background ODV decreases with distance both east and south across the state, with the smallest values found in southern and eastern Texas in the Southwest Texas (50 ± 5 ppb) and Houston (54 ± 3 ppb) regions, which are heavily influenced by maritime air from the Gulf of Mexico (Berlin et al., 2013) during the warm (ozone) season. This broad geographic feature in background ozone is also seen in several modelling studies (e.g., Fiore et al., 2014; Zhang et al., 2020).

As expected, the largest Texas US anthropogenic ODV enhancements (27 to 54 ppb in 2000) have generally been found in the nonattainment areas within the state - Houston, Dallas and San Antonio – although comparably large $A$ parameter values (28 to 37 ppb) are found throughout southeast and east Texas. It is notable that the El Paso region is anomalous – although the $A$ parameter value (11 ± 2 ppb) is comparatively modest, the $a$ parameter value (65 ± 2 ppb) is so large that even the relatively small US anthropogenic ODV enhancement is sufficient to produce ODVs that routinely exceed the 70 ppb NAAQS throughout the entire measurement record (upper left graph in Figure 8).

There is evidence that the maximum ODVs in Texas urban areas may be elevated by ozone sources that have not yet been effectively controlled by air quality improvement efforts. Such sources could possibly include wildfire emissions, which are approximately quantified by Equation 4, or emissions from oil and gas exploration and production, which are ubiquitous across Texas. To gauge the possible magnitude of these sources, Equation 4 is fit to the time series of maximum ODVs in Dallas and Houston (dotted black curves in the middle graphs of Figure 8). These fits possibly indicate contributions of 2 to 3 ppb to the maximum ODVs in these urban areas from such growing, uncontrolled anthropogenic sources. Iglesias et al. (2022) show the



significant growth in wildfires across all of Texas (their Figure 2), especially since the turn of the century, and Gong et al.
(2017) calculate that wildfires impact Houston on 3.5% of the days in their May-September 2008-2015 study period, with an average contribution of ~8.5 ppb.

### 4.5 ODV contributions in neighboring and more distant states

For context, preliminary analyses have been conducted of the ODV time series recorded in states lying east and north of the region that is the focus of this study. Figures S4 and S5 illustrate the results of these analysis for Oklahoma, Louisiana,
Arkansas, Nebraska and Kansas; Equation 3 is fit to the time series of all ODVs recorded in each state without consideration of intrastate regional differences. Nevertheless, the RMSD of the ODVs about the fits are similar to the fits to more carefully selected sets of sites discussed in the preceding sections. The results for these states are considered to be adequate for determining boundary conditions for the ODV component determinations in the region that is the focus of this work.

Figure S4 also includes analyses of ODVs recorded in four rural northern states: Montana, North and South Dakota, and
Wyoming. A fit to all ODVs recorded in these four states gives a mean $A$ parameter value of $1.3 \pm 1.7$ ppb, which indicates only small US anthropogenic ODV enhancements throughout that region. (Note: the Boulder site in Wyoming – AQS Site ID 56-035-0099 reports anomalously large ODVs; since this site is in the Upper Green River Basin where high ozone concentrations are observed in wintertime (Schnell et al., 2009), ODVs from this site have been excluded). Separate fits of Equation 3 to the ODV values in each state provide separate $a$ parameter values; there are some small differences in these
results: a maximum in Wyoming of 64 ppb, a minimum in Montana (59 ppb), with North and South Dakota being intermediate (60 and 62 ppb, respectively). Parrish and Ennis (2019) present an earlier analysis for three of these states, and found results consistent with this updated analysis.

Figure S7 includes analyses using fits of Equation 4 to the time series of maximum ODVs recorded in the New York City and Atlanta urban areas. The New York City time series was originally analyzed by Parrish and Ennis (2019) (see their Figure
S4); here that time series is extended through 2021. Figure S7 includes separate symbols for the Atlanta, coastal and inland sites.

### 5 Discussion and Conclusions

### 5.1 Background ozone dominates observed concentrations throughout the US

The left graphs of Figure S1 compare five estimates of the May to September US background ozone distribution derived
from analysis of observations from CA West Coast locations. The first bar illustrates the ozone distribution measured between 0.6 and 1.0 km altitude by sondes launched from a coastal site at Trinidad Head (Oltmans et al. 2008), which is located in the Northern Coast AB shown in map in Figure S1; Parrish et al. (2022) fully describe this data set. This altitude range is selected to be high enough to avoid continental influences as air is transported ashore and low enough to represent air mixed into the convective boundary layer over the continent. Thus, these sonde data provide direct estimates of US background ozone

concentrations. The other four bars show estimates of the distribution of US background ozone derived in Section 4.2 from the MDA8 distributions in CA coastal air basins. There is reasonable agreement between the five estimates, but with some notable differences. The two southern estimates (San Diego AB and SoCAB) are somewhat larger than the three more northerly, which is consistent with the systematic decrease of US background ozone with latitude quantified by Parrish et al. (2022). Variation in site elevations also contributes to these differences. The Crestline site at 1.4 km and the Alpine site at the 0.6 km are included

in the SoCAB and San Diego AB, respectively; they are at higher elevations than other coastal air basin sites, and very often record the largest MDA8 ozone concentrations in their respective AB. Thus, at least some of the quantitative differences in the bars in the left graphs of Figure S1 represent real atmospheric differences. The mean of all five background ozone distributions from the lower graph is included in Figure 3; it provides an estimate for the average US background ozone distribution along the CA coast.

Comparison of the distributions in Figures 3 (left graphs) and S1 shows that by 2015 the distribution of US background ozone was larger than the distribution of the US anthropogenic ozone enhancements in all four CA coastal air basins. This dominance of background ozone is quite pronounced; only in the SoCAB and San Diego AB do the two ozone distributions in Figure S1 overlap at all; in the SoCAB the 75[th] percentile of the US anthropogenic ozone enhancement is smaller than the 10[th] percentile of the background contribution. The lower graphs of Figure S1 show the comparisons for the separately derived

air basin background ozone distributions; here the US background ozone dominance in the SoCAB and San Diego AB comparisons are even more pronounced. This predominance of the background contribution developed relatively recently; the upper graphs of Figure S1 show the comparison for the year 2000, when the US anthropogenic enhancements were generally greater than or equivalent to the background contributions in the SoCAB, and more nearly comparable in the San Diego AB.

The predominant contribution of US background ozone to maximum ozone concentrations, even in the SoCAB where the

largest US ozone concentrations occur, emphasizes the great difficulty of achieving accurate air quality modeling. Traditionally, CTMs have primarily focused on careful, detailed treatment of transport and photochemical processes on local to regional scales, which is required for the accurate simulation of the US anthropogenic ozone enhancements; however, less attention has been paid to the US background ozone contributions, especially the day-to-day variability. Of great importance for present day exceedance event modeling or daily air quality forecasts, is the accurate simulation of the US background

ozone on short-term, i.e., daily or even hourly, time scales. Since this ozone contribution is driven by a manifold of processes occurring on small spatial scales, but over hemisphere-wide distances, accurate, detailed simulation of the stratospheric and tropospheric contributions to US background ozone is beyond the capabilities of current air quality modeling, as is such modeling for predictions of future baseline ozone concentrations. These issues are explored in two recent comparisons of estimates of background ozone and their impacts on the simulation of surface ozone across the continental US (Hogrefe et al.,

2018; Zhang et al., 2020) where background concentrations were shown to differ by 5-10 ppb in the mean and up to 15 ppb for domain-averaged MDA8 values. The graph at right in Figure 3 illustrates the differences between two different model simulations of the US background MDA8 ozone distributions for three months at high elevation sites in the western US. As Zhang et al. (2020) note, the means of the distributions agree within 6 ppb. However, the larger percentiles of the distribution



differ more widely; e.g., the 98$^{th}$ percentiles are ~70 ppb and ~57 ppb from the GFDL AM4 and GEOS Chem models,
respectively. It's notable that the former model indicates that US background ozone alone can exceed the 2015 NAAQS of 70
ppb, while the latter model finds that US background ozone always remained significantly below the NAAQS.

## 5.2 Pronounced shift in spatial distribution of maximum US ozone concentrations

Great efforts have been made over the past half century to reduce anthropogenic precursor emissions throughout the entire
country. These efforts have been remarkably successful, yielding substantial reductions in the number and magnitude of US
ozone NAAQS exceedances; however, progress has not been spatially uniform over the country. Figure 9 illustrates one
measure of this progress. During the 5-year period of 1997-2001 the large majority (43) of the 51 US states plus the District
of Columbia recorded at least one ODV of 75 ppb or larger; 20 years later, during the 2017-2021 period, the number of such
states had been reduced to 20. In the earlier period 80% of ODVs across the Nation were ≥75 ppb, and in the later period that
percentage had dropped to 10%, clearly demonstrating a remarkable improvement in ozone air quality. However, there are
marked regional differences in the percentages of ODVs ≥75 ppb - 88% and 64% in the eastern and western regions,
respectively, in the early period, but 2.4% and 22%, respectively in the later period. In Figure 9 the solid blue bars in the
western region demonstrate that nearly all of the seven total states that recorded more than 7% of ODVs ≥75 ppb were in the
southwestern US, including Texas and California; the single exception is Connecticut, where the maximum ODVs are recorded
near the coast of Long Island Sound (Parrish and Ennis, 2019). Similar conclusions are reached if ODVs > 70 ppb are
considered (Figure S6). In summary, great improvement in US ozone concentrations have occurred over past decades, but that
improvement has been accompanied by a marked shift of the highest observed ozone concentrations from the eastern to the
southwestern US.

The occurrence of only modest ODV decreases in the southwestern US has been noted in other analyses. Simon et al.
(2015) found that the upper end of the summertime urban ozone distribution (i.e., representative of ODVs) generally decreased
over the 1998-2013 period throughout the US in response of decreasing NO$_X$ and VOC emissions. They interpreted those
results as demonstrating the large scale success of US control strategies. However, their Figure S12 shows that the Denver
urban area was an exception to this generality, with all sites exhibiting either positive or insignificant trends. Similarly, Abeleira
and Farmer (2017) found that ozone in the Denver urban area was either stagnant or increasing between 2000 and 2015, despite
substantial reductions in NO$_X$ emissions. In Colorado Springs, approximately 100 km south of Denver, Flynn et al. (2021)
found that summertime MDA8 ozone shows no significant trend throughout its distribution over the past 20 years. Further,
they found little evidence of local photochemical production or ozone transport from Denver, consistent with our classification
of this area as rural. The results of these three studies are consistent with the long-term ODV changes in urban and rural
Colorado shown in Figure 5, and our conclusion that the US background ODV is the dominant contributor to ODVs.





### 5.3 Spatial and temporal distribution of ODV contributions

The derived spatial distributions of the two major ODV components – (1) the US background ODV, and (2) the US anthropogenic ODV enhancement – are illustrated for the year 2000 in the two contour maps in Figures 10 and 11. In Figure 10 the estimated US background ODVs are at a maximum - greater than 64 ppb (see violet contour) - throughout the entire southwestern US, including western Texas and Wyoming, and above 68 ppb in some areas, including most of Colorado. We attribute these large values to baseline air at higher altitudes flowing over the inland mountain ranges, and mixing into the

convective boundary layer over the continent (Langford, et al., 2017). The overall spatial pattern is similar to that simulated for surface impacts of ozone transport from Asia (see Fig. 9 of Lin et al., 2012a), stratospheric intrusions (see Fig. 11 of Lin et al., 2012b) and US background ozone distributions (see Fig. 1 of Zhang et al., 2020). Previous studies of reanalysis data sets (Sprenger and Wernli, 2003; Škerlak et al., 2014) describe the southwestern US as prone to high rates of deep stratosphere to troposphere transport. These events tend to cross the tropopause at high latitudes (40-60°N) near the end of the North Pacific

storm track but make their surface impacts felt most acutely after quasi-isentropic subsiding transport to the southeast (in the lee of the Pacific High). While the peak of this effect occurs in the spring it does continue into the summer (Škerlak et al., 2014), when it can more strongly couple with the surface because of the very deep convective boundary layers that develop in the arid, high terrain of the southwestern US. With such large US background ODV values, even relatively small anthropogenic enhancements in the regional urban areas raise ODVs to values larger than the 70 ppb ozone NAAQS, thereby leading to their

designation as nonattainment areas. A striking feature of the contour map in Figure 10 is the significant spatial gradient in the US background ODV over the continent. Increases occur inland from the West Coast as surface elevations increase (Parrish et al. 2022), with decreases to the north and east of the broad maximum in the southwestern US, and to the east and south within Texas. These smaller values in eastern Texas (e.g., 54 ± 3 ppb in Houston) allow for larger anthropogenic ODV enhancements in that region before the NAAQS is exceeded.

CTMs have been extensively utilized to estimate US background ODVs. Figure 10 compares our observation-derived results with a portion of Figure 3 of Jaffe et al. (2018), which presents the results of a simulation by the GFDL-AM3 model of a statistic comparable to the US background ODV. The broad spatial features in both plots are similar; the dominant feature in each is the maximum in the southwestern US. The maximum observation-derived values (68-70 ppb) are consistent with the model simulated maximum (60-70 ppb). All of the observation-derived values, except for a narrow band along the Pacific

Coast, are in the 50 to 70 ppb range, as are the model derived values, with the exception of the values in the southeast corner of that figure. Both results show general south-to-north and west-to-east decreases of inland US background ODVs. However, it is apparent that the observational-based estimate has substantially greater precision and spatial resolution than provided by the low resolution (200 x 200 km²) model estimate. Jaffe et al. (2018) estimate that the model uncertainty of the seasonal mean US background ozone is ~ ±10 ppb, but larger for individual days, as is required to determine US background ODVs. The

magnitudes of the model and observational-derived results in Figure 10 agree within this estimated uncertainty of the model simulations. The US background ODV estimated from the GFDL-AM4 simulated ozone distribution (approximated by the



98[th] percentile) shown in Figure 3 is also consistent with the contour map in Figure 10, indicating that US background ODVs above 68 ppb over the southwestern US are indeed physically realistic.

The $NO_2$ column measurements from the OMI satellite (Lu et al., 2015) are included in Figure 11 for comparison with the US anthropogenic ODV enhancements, which outside CA, are generally elevated in the larger southwestern US and Texas urban areas. There is general correlation of the $NO_2$ columns with these urban areas, which supports our interpretation of the derived *A* parameter value as quantifying the US anthropogenic ODV enhancement. However, some areas of clear differences between those ODV enhancements and the OMI $NO_2$ column measurements are evident in Figure 11. Most clearly, emissions from the large coal-fired power plants in the Four Corners region (location identified in Figure S2) give a strong elevation in the $NO_2$ column that is not reflected in US anthropogenic ODV enhancements. This lack of correlation is consistent with the ODV time series for this region (Figure S3), which indicates that the US anthropogenic ODV enhancement is near zero. Mesa Verde NP is in this region; it has been considered a high-elevation site representing baseline $O_3$, at least in the spring, by Lin et al. (2017). This site records ODVs generally consistent with all other sites in this region. The co-location of the large $NO_2$ enhancement with indiscernible US anthropogenic ODV enhancement indicates that the large $NO_2$ emissions from the power plants have a negligible effect on the regional ODVs. The Denver urban area, with *A* parameter values in the 8 to 11 ppb range (Figure 5), does not stand out in Figure 11, although that area does have a clear signature in the OMI $NO_2$ column data.

Figure 12 compares the long-term changes of the ODV components between the Crestline site in the Los Angeles urban area with those in the four larger southwestern US urban areas and three Texas urban areas. Since ~1990, the Crestline site has usually recorded the largest ODV in the SoCAB. Overall, decreases in ODVs have been recorded in each of the eight cities; this decrease is primarily driven by a decrease in the US anthropogenic ODV enhancement. The final term in Equation 3 indicates that this contribution decreased by a factor of 6.3 between 1980 and 2020. In contrast, the first 3 terms of Equation 3 indicate that the US background ODV increased over the entire western US region by ~11 ppb from 1980 to the mid-2000s, and has since slowly decreased. The relatively minor, but increasing (up to ~ 6 ppb in Denver in the year 2020) mean ODV enhancements due to wildfire emissions quantified in five urban areas are separately indicated. Two conclusions emerge from Figure 12. First, throughout the monitoring record, the US Background ODV has been, by far, the largest ODV contribution in the four southwestern US urban areas, and that by 2000 it had become the predominant ODV contribution in all cities, even within the Los Angeles urban area exemplified by the Crestline site. This conclusion is consistent with the discussion in Section 5.1. Second, the increase in the US background ODV between 1980 and the mid-2000s and the increasing wildfire contribution through the entire period, partially offset the reduction in the US anthropogenic ODV enhancement, so that ODVs have not changed greatly in the southwestern US urban areas and El Paso, Texas.

There are some notable deviations from the overall picture described above. First, the results shown in Figure 12 and all fitted curves in all figures represent the average behavior of the fitted ODVs; there is significant scatter about those fits as quantified by the RMSDs (generally < 5 ppb) annotated in the figures and given in Tables 1, S1 and S3-S5; this scatter appears to be predominately year-to-year variability. However, one systematic deviation is particularly relevant - the recent anomaly of the El Paso nonattainment area and nearby rural New Mexico (see graphs in Figures 8 and S3); the ODVs in the last few



years are as much as 15 ppb larger than expected from the fits to Equation 3. It is notable that in the most recent year included in this analysis (2021) the largest ODV (80 ppb) in the two state region of Texas and New Mexico was not recorded in either of the traditional urban ozone hot spots of Houston or Dallas; rather it was recorded at the rural Desert View site in New Mexico, which is included in the El Paso region in Figure 8. Only a single site in Phoenix within the entire southwestern US

and Texas region (excluding California) equaled that ODV. ODVs nearly as high were recorded in rural New Mexico sites that are included in the Western Rural TX region in Figure S3. The cause of these high ODVs is presently unexplained, although it may be important to note that these regions are in the Permian oil and gas basin (Karle et al., 2021).

### 5.4 Implications for US air quality policies

This spatial difference in the success of air quality policies aimed at ODV reduction has resulted in a pronounced regional

shift of the occurrence of ODVs of 75 ppb and above (and above 70 ppb – Figure S6), with a growing preponderance of the nation's largest ODVs recorded in the southwestern US (see discussion in Section 5.2). The cause of this change is clear in Figure 10: the southwestern US suffers from large US background ODVs that closely approach the NAAQS. In Figure 12, the US anthropogenic ODV enhancements in the southwestern US urban areas and El Paso Texas have been reduced in magnitude to such an extent that by 2020 all were smaller than 6 ppb. However, even these remaining enhancements are large enough to

raise the recorded ODVs above the 70 ppb 2015 NAAQS, as well as the 75 ppb 2008 NAAQS. This conclusion is further illustrated in Figure 13, where fits of Equation 4 to time series of maximum ODVs in three southwestern US urban areas are compared to similar fits in three of the largest US urban areas; additionally, the fit of Equation 3 to the time series of CASTNET ODVs and the temporal evolution of the corresponding baseline ozone concentrations from Figure 1 are included. In Atlanta and New York City, with substantially lower year 2000 maximum US background ODVs of 49 and 52 ppb, respectively, the

fits to the urban maximum ODVs dropped below the Phoenix and Denver fits in the early 2010s; they have now dropped below the 70 ppb NAAQS and are approaching the Reno, CASTNET and baseline fits. Interestingly, by 2021, the fit to the maximum ODVs recorded at all locations in Figure 13, from the isolated rural CASTNET sites in the western US to the major metropolitan areas of Atlanta and New York City (excluding the Los Angeles) were in the 66 to 76 ppb range, with the southwestern US urban areas defining the upper limit. Importantly, all curves included in Figure 13 are direct fits to measured

concentrations; they do not depend upon accurate differentiation between the separate ODV contributions.

The spatial shift of the nation's highest ozone concentrations to the southwestern US emphasizes a long-standing concern with US policies for improving ozone air quality – a single standard is applied to the entire country without regard to the ozone background; this background cannot be addressed by local or regional reductions in ozone precursors. This policy contrasts to those addressing another environmental hazard – exposure to ionizing radiation. The U.S. Nuclear Regulatory Commission

(NRC, https://www.nrc.gov/about-nrc/radiation/around-us/doses-daily-lives.html) estimates that about half of the average American radiation dose is due to natural background radiation that, similar to ozone, is higher in the high altitude southwestern US. Health and safety standards for ionizing radiation are based on the additional exposure that comes from anthropogenic sources of radiation, not on the overall radiation exposure. A similar approach for ozone would require that the ozone standard





be based on the anthropogenic increment of ozone above the US background ozone within a region, rather than on the total

ambient ozone concentration. Alternatively, a regionally varying ambient concentration standard that accounted for the regionally varying US background ozone could serve that purpose. As of 2020, we estimate that the US anthropogenic ODV enhancement in eight southwestern US urban areas (seven in Figures 4, plus El Paso TX) had been reduced to the range of 2.4 to 6.4 ppb; yet the US background ODVs, plus the relatively small wildfire contributions, are so large that 5 of these 8 cities are the centers of ozone nonattainment areas. Importantly, it is only the small US anthropogenic ODV enhancement that can

possibly be directly reduced through further controls on urban and industrial emissions, although in addition the small wildfire contributions possibly can also be reduced indirectly through decreased urban $NO_X$ concentrations. In Denver in 2020, we estimate that the US anthropogenic ODV enhancement had been reduced to 4.4 ppb, yet the US EPA has recently downgraded the Denver urban area from a "Serious" to "Severe-15" nonattainment area under the 2008 ozone NAAQS (https://www.govinfo.gov/content/pkg/FR-2022-10-07/pdf/2022-20458.pdf). This redesignation will require further

reductions in local and regional precursor emissions even though the US anthropogenic ODV enhancement is already so small that such additional emission control efforts can be expected to have little impact on future ODVs.

Importantly, it is estimated that between 1950 and 2000 baseline ozone concentrations at northern midlatitudes increased by a factor of 2.1 ± 0.2 (Parrish et al., 2021b). These baseline concentrations, which largely account for the US background ODV, reached a maximum in the mid-2000s, and since have slowly decreased at a mean rate of ~1 ppb decade$^{-1}$ (Parrish et

al., 2020). This decrease results from decreasing anthropogenic ozone precursor emissions throughout northern mid-latitudes due to implementation of effective emission controls in North America, Europe and, more recently, east Asia. Given the large increase in baseline ozone that occurred as anthropogenic emissions increased in the 20$^{th}$ Century, it can be expected that a significant decrease in baseline ozone can be achieved by further reducing the total zonal precursor emissions. Cooperative, international emission control efforts aimed at continuing or even accelerating the current decrease of baseline ozone

concentrations, and thereby the US background ODV, may be an effective alternative policy approach to further reducing US ODVs. It should be noted that his expectation relies on the assumption that the background will continue to fall; however that expectation could be compromised by the changing climate. For example, stratosphere-troposphere exchange rates could increase due to acceleration of the Brewer-Dobson circulation; modeling work by Abalos et al. (2020) points toward a 10-15% rise in the stratosphere-troposphere $O_3$ source by the end of this century.

Finally, we should step back from comparison of air quality policy approaches to note that there is evidence that further reductions of ambient ozone throughout the US will provide significant health benefits (e.g., Zhang et al., 2019). This applies to both attainment and nonattainment areas, and to the entire concentration distribution of ozone, not just the few days of highest concentrations that determine the ODVs that are the focus of NAAQS attainment. Thus, all reductions of ozone precursor emissions will be beneficial, and if carried out in all northern midlatitude countries, will serve to continue reductions

in baseline ozone concentrations, thereby easing the attainment of air quality standards based on ODVs or other statistics designed to quantify the highest ozone concentrations.



**Acknowledgments, Samples, Data and Legal Notice**

D.D. Parrish is an independent consultant (David D. Parrish, LLC); his effort was supported by the Coordinating Research Council, Inc. (CRC) through Contract No. A-129. The Coordinating Research Council, Inc. (CRC) is a non-profit corporation supported by the petroleum and automotive equipment industries. CRC operates through the committees made up of technical experts from industry and government who voluntarily participate. The four main areas of research within CRC are: air pollution (atmospheric and engineering studies); aviation fuels, lubricants, and equipment performance; heavy-duty vehicle fuels, lubricants, and equipment performance (e.g., diesel trucks); and light-duty vehicle fuels, lubricants, and equipment performance (e.g., passenger cars). CRC's function is to provide the mechanism for joint research conducted by the two industries that will help in determining the optimum combination of petroleum products and automotive equipment. CRC's work is limited to research that is mutually beneficial to the two industries involved. I.C. Faloona's effort was supported by the USDA National Institute of Food and Agriculture, (Hatch project CA-D-LAW-2481-H, "Understanding Background Atmospheric Composition, Regional Emissions, and Transport Patterns Across CA").

**Competing Interests**

The contact authors have declared that none of the authors has any competing interests.

**Author contributions**

D.D.P. was responsible for the overall design. I.C.F. provided analysis. D.D.P. wrote the paper with input from I.C.F. and R.G.D. All authors: edited and revised manuscript.

**Data availability statement**

The California MDA8 ozone concentrations were obtained from the CA Air Resources Board archive (https://www.arb.ca.gov/adam/index.html; last accessed 24 October 2023). The ODVs were obtained from EPA's AQS data archive (https://www.epa.gov/aqs; last accessed 17 June 2022).

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





**Table 1.** Parameter values (with 95% confidence limits) derived from fits of Equation 1 to time series baseline ozone concentrations. Represents the coefficient of the cubic term in the analogous fit to a third order polynomial.

| Data Set | $b$ (ppb yr$^{-1}$) | $c$ ($10^{-2}$ ppb yr$^{-2}$) | year$_{max}$ | RMSD (ppb) | Reference | $d$ ($10^{-4}$ ppb yr$^{-3}$) |
|---|---|---|---|---|---|---|
| 2-year means of baseline ozone | $0.20 \pm 0.06$ | $-1.8 \pm 0.6$ | $2005.7 \pm 2.5$ | 1.4 | Parrish et al., 2020 | $+0.6 \pm 5.7$ |
| CASTNET ODVs – normalized[1] | $0.07 \pm 0.13$ | $-1.5 \pm 0.8$ | $2002.1 \pm 4.4$ | 2.4 | This work | $+5.8 \pm 10.1$ |

[1] The $a$ parameter value derived from the fit of Equation 1 to the normalized CASTNET ODVs is $71.3 \pm 0.8$ ppb, which is consistent with the normalization process utilized here; the corresponding value in the Parrish et al. (2020) analysis was near zero due to the different normalization approach used in that work.




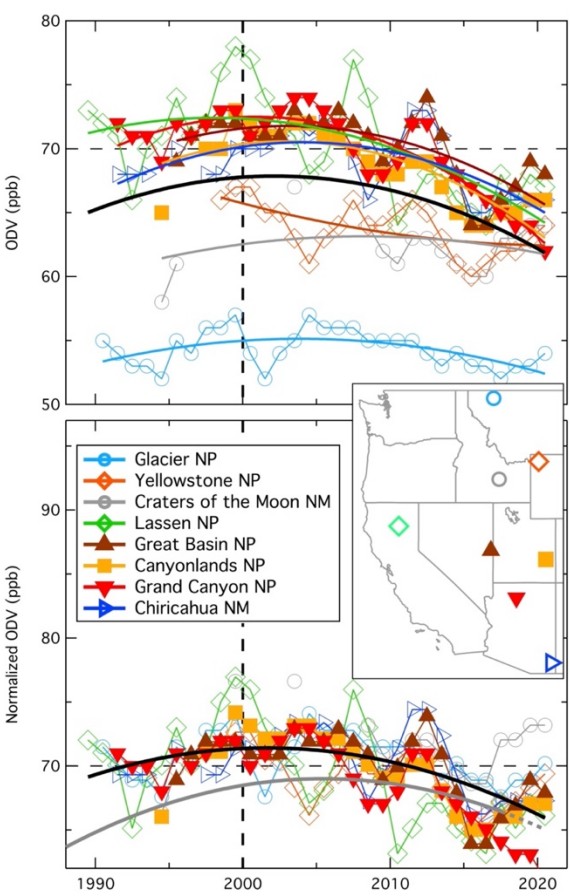

**Figure 1:** Time series of ODVs recorded at the eight rural western CASTNET sites shown in the inset map. Solid symbols indicate the three sites used for the normalization. Solid curves indicate the fits of Equation 1 to the individual site data (upper panel, including the black curve fit to all data) and to all normalized data (black curve, lower panel). The lower grey curve in the lower panel is the Parrish et al. (2020) fit to baseline data, normalized to 68.5 ppb (the *a* parameter value derived in a fit of Equation 3 to the CASTNET data) in the year 2000; dashed line shows extrapolation of that fit to 2021.



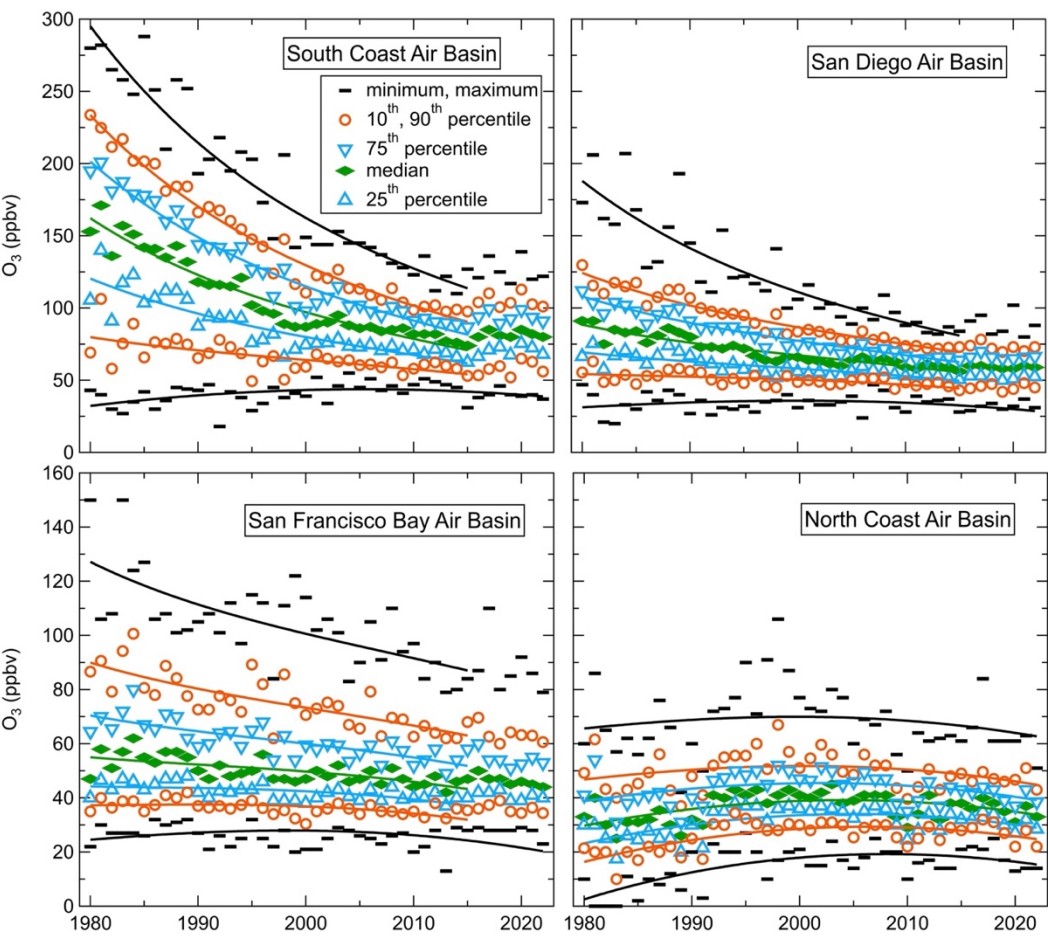

**Figure 2:** Temporal evolution of maximum MDA8 ozone concentrations in four California air basins. Symbols indicate the annotated percentiles of the annual distribution of the maximum MDA8 concentration recorded at any monitor within the basin on each day of the May–September ozone season. Solid curves indicate fits of Equation 3 to the respective percentiles.



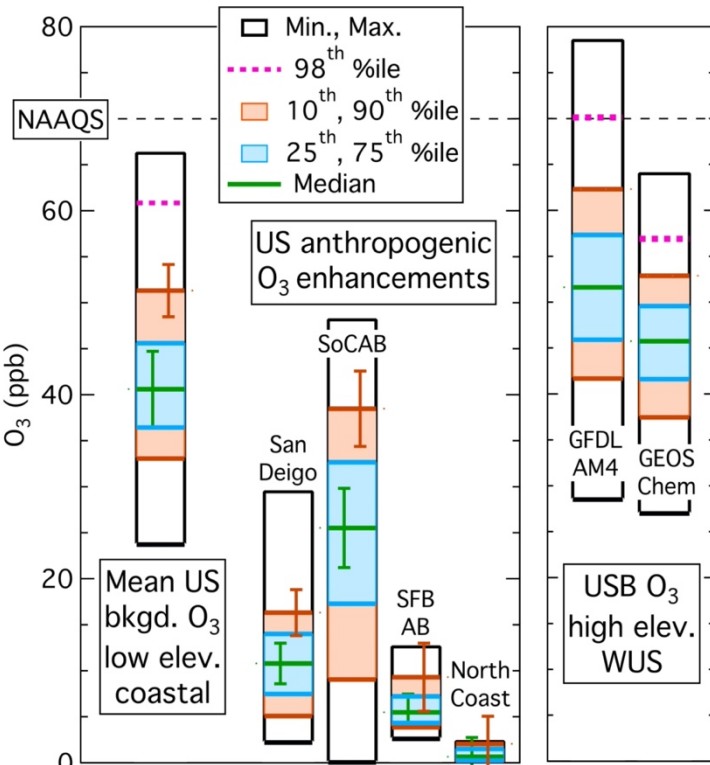

**Figure 3. (left)** Comparison of year 2015 mean distribution of US background ozone with the distributions of US anthropogenic ozone enhancements in four coastal CA air basins. Error bars on the median and 90th percentile lines indicate standard deviations of the means from the five background determinations and the estimated uncertainties of the US anthropogenic ozone enhancements. **(right)** Distribution of daily MDA8 US background ozone for April–June 2017 at 12 western US high-elevation sites simulated with GFDL-AM4 and GEOS-Chem global models; these bar graphs were derived from the plots in Figure 17 of Zhang et al. (2020).



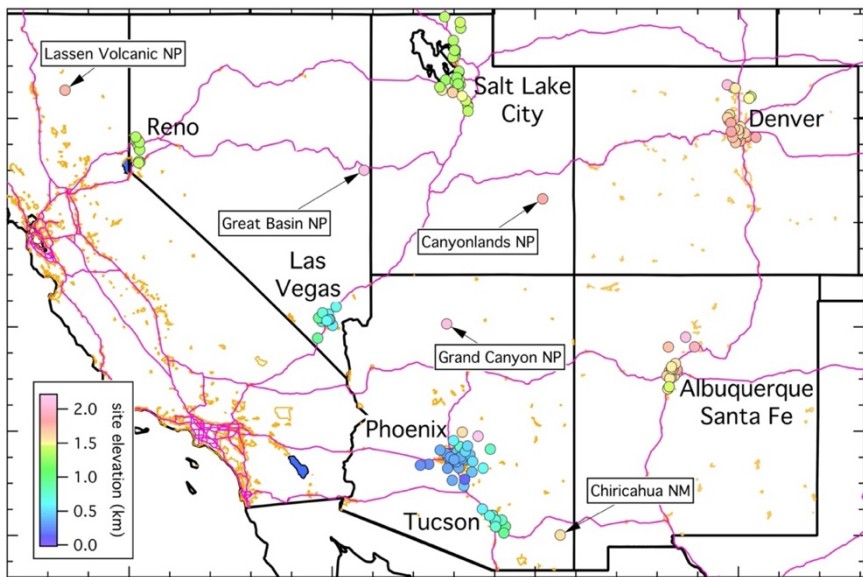

**Figure 4:** Map of southwestern US urban monitoring sites; the symbols are color-coded according to site elevation as indicated in the annotation. Lines indicate outlines of southwestern US states (black), urban areas (gold) and interstates and selected other major highways (violet). Seven urban areas, whose sites are analyzed together as separate data sets, are labelled. Locations of five of the isolated rural CASTNET sites are also included.

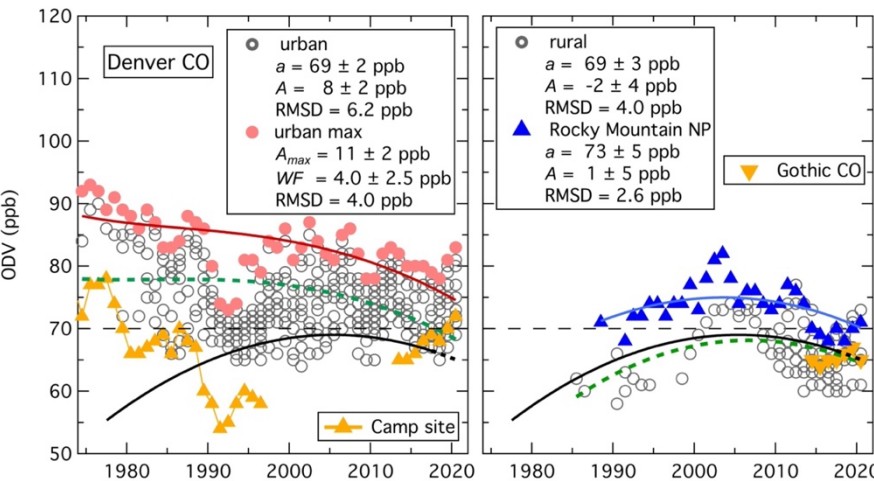

**Figure 5:** Time series of ODVs recorded in Colorado at the urban sites (left graph) whose locations are shown in Figure 4, and at rural sites (right graph, excluding the Four Corners area) whose locations are shown in Figure S1. Open symbols indicate all ODVs recorded in each area, and solid symbols indicate ODVs from three specific sites and the maximum Denver urban ODVs in each year. Green dashed curves indicate fits of Equation 3 to all plotted ODVs (excluding the Camp and Rocky Mountain NP sites). Red solid curve on left indicates fit of Equation 4 to the maximum Denver urban ODVs and blue curve on right indicates Rocky Mountain NP fit of Equation 3; parameters derived in these fits are annotated in the graphs. The black solid curves with dashed extensions indicate the fit to the baseline data included as the grey curve in Figure 1. The light dashed lines indicate the 70 ppb ozone NAAQS.







**Figure 6:** Time series of ODVs recorded in six southwestern US urban areas shown in Figure 4. Open symbols indicate all ODVs recorded in each area, and the solid symbols indicate the maximum ODVs in each year in each area. Green dashed curves indicate fits of Equation 3 to all ODVs; red solid curves indicate fits of Equation 4 to the maximum ODVs, with the parameters derived in this fit annotated. The black solid curves with dashed extensions indicate the fit to the baseline data from Figure 1, normalized to the respective *a* parameter values. The light dashed lines indicate the 70 ppb ozone NAAQS.





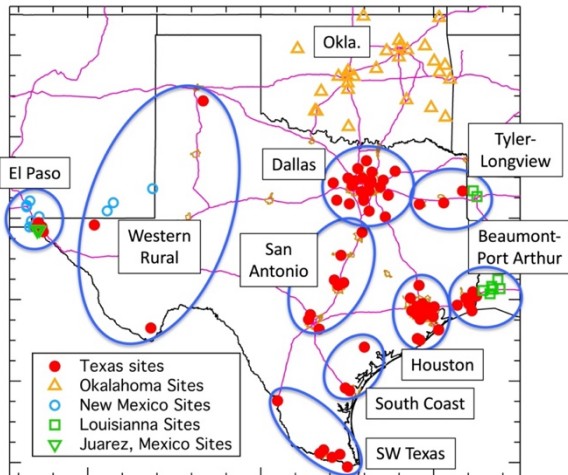

**Figure 7:** Map of monitoring sites within the Texas region; the symbols are color-coded according to state as indicated in the annotation. Lines indicate outlines of southwestern US states (black), urban areas (gold) and interstates (violet). Nine Texas regions, whose sites are analyzed together as separate data sets, are indicated.





**Figure 8:** Time series of ODVs recorded in seven of the Texas regions shown in Figure 7. Grey symbols in each graph indicate all recorded Texas ODVS. Colored symbols indicate the ODVs from each respective region. Upper solid curves indicate fits of Equation 3 to all ODVs in the area over the curve's time span; the parameters derived in these fits are annotated in three graphs; dotted curves indicate fits of Equation 4 to the maximum ODVs in El Paso, Houston and Dallas, with the parameters derived in these fits annotated. Lower solid curves with dashed extensions indicate the fit to the baseline data from Figure 1, normalized to the respective *a* parameter values. The light dashed lines indicate the 70 ppb ozone NAAQS.



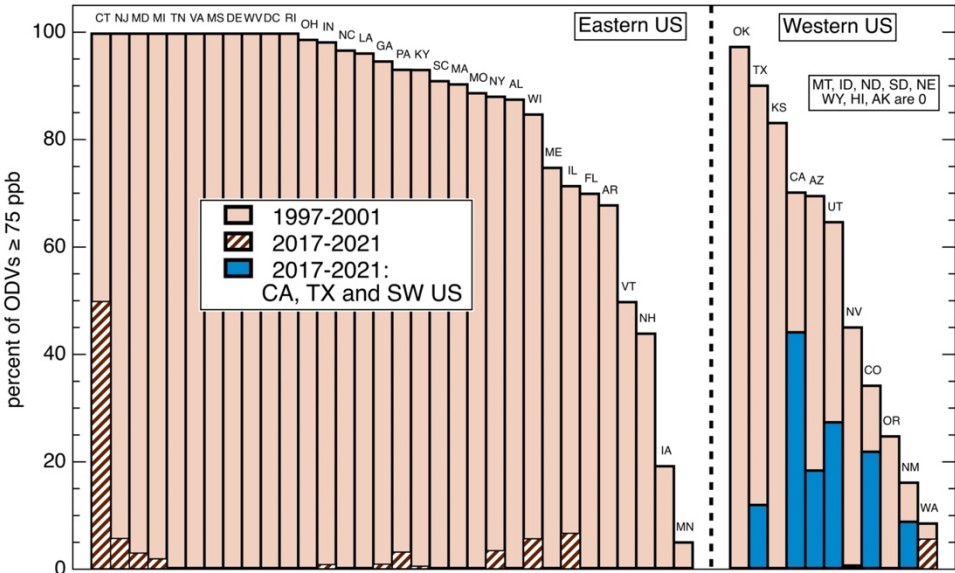

**Figure 9:** Comparison of percentage of ODVs greater than or equal to 75 ppb recorded at all sites in individual states over two 5-year periods: 2017-2021 (hatched and dark blue bars) and a period 20 years earlier - 1997-2001 (light-colored bars). Individual states are indicated by their two letter abbreviations (defined in Table S6). States are arbitrarily divided between eastern and western regions. Southwestern states, Texas and California are indicated by solid dark blue bars. Eight rural states, all in the western region, reported no ODVs greater than or equal to 75 ppb in either period.



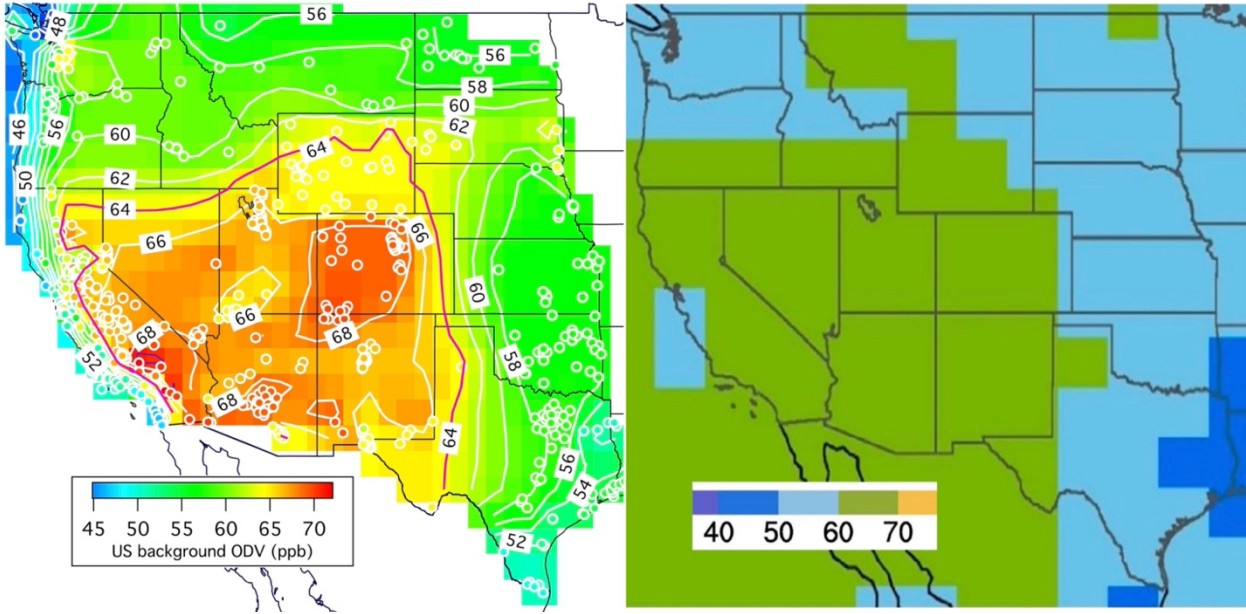

**Figure 10:** Two estimates of the spatial variability of US background ODVs over the western US. **(Left)** Contour map of estimated US background ODV for year 2000. The US background ODV excludes estimated wildfire contributions. The 64 ppb contour is colored violet to indicate the extensive area of largest values. Symbols in the contour map indicate individual monitoring sites included in the analysis with the same color coding as the contour map. Results from Parrish et al. (2017; 2022) and estimates for California's Central Valley (Faloona et al., manuscript in preparation) are included. **(Right)** Annual 4th highest MDA8 ozone concentration averaged over 2010–2014, from a GFDL-AM3 model simulation with North American anthropogenic emissions zeroed out (figure reproduced from a section of Figure 3 of Jaffe et al., 2018).



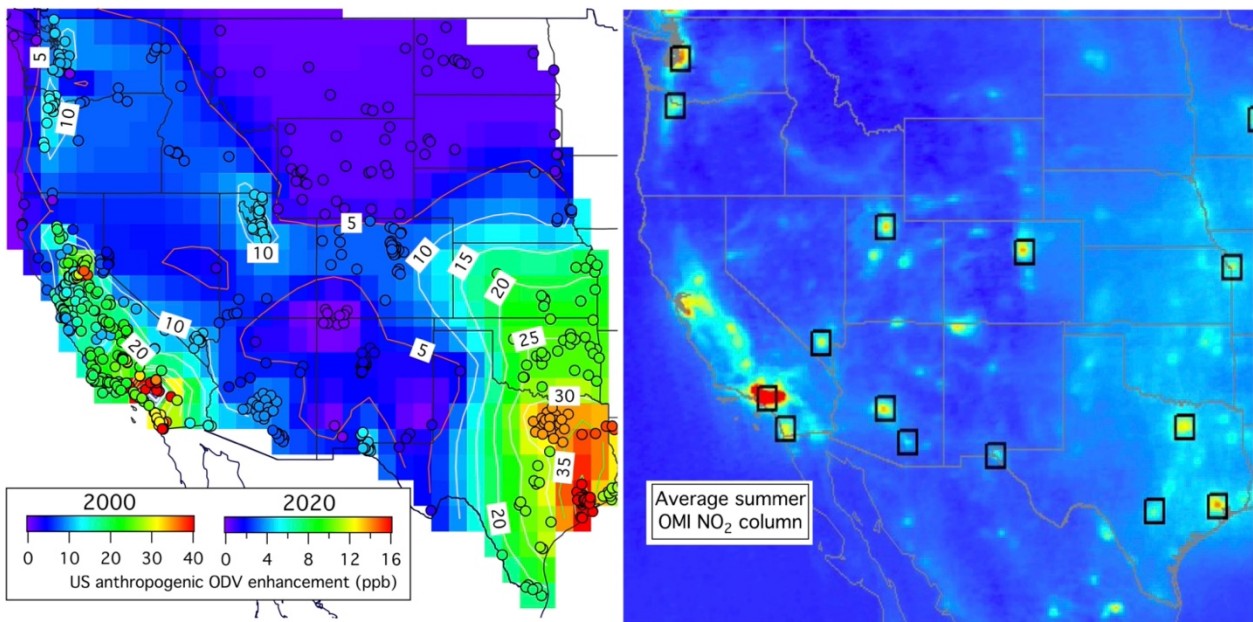

**Figure 11:** US anthropogenic ODV enhancement over the western US compared to mean summertime OMI NO$_2$ columns. **(Left)** Contour map of estimated US anthropogenic ODV enhancement for year 2000 over the western US. Symbols in the contour map indicate individual monitoring sites included in the analysis with the same color coding as the contour map. A second color scale is included for interpretation of the colors as the year 2020 US anthropogenic ODV enhancement. Results from Parrish et al. (2017; 2022) and estimates for California's Central Valley (Faloona et al., manuscript in preparation) are included. **(Right)** The OMI NO$_2$ columns measured in April – September 2005-2014 are reproduced from Figure 1a of Lu et al. (2015) by cropping and changing the aspect ratio of that figure to approximate that of the contour map.





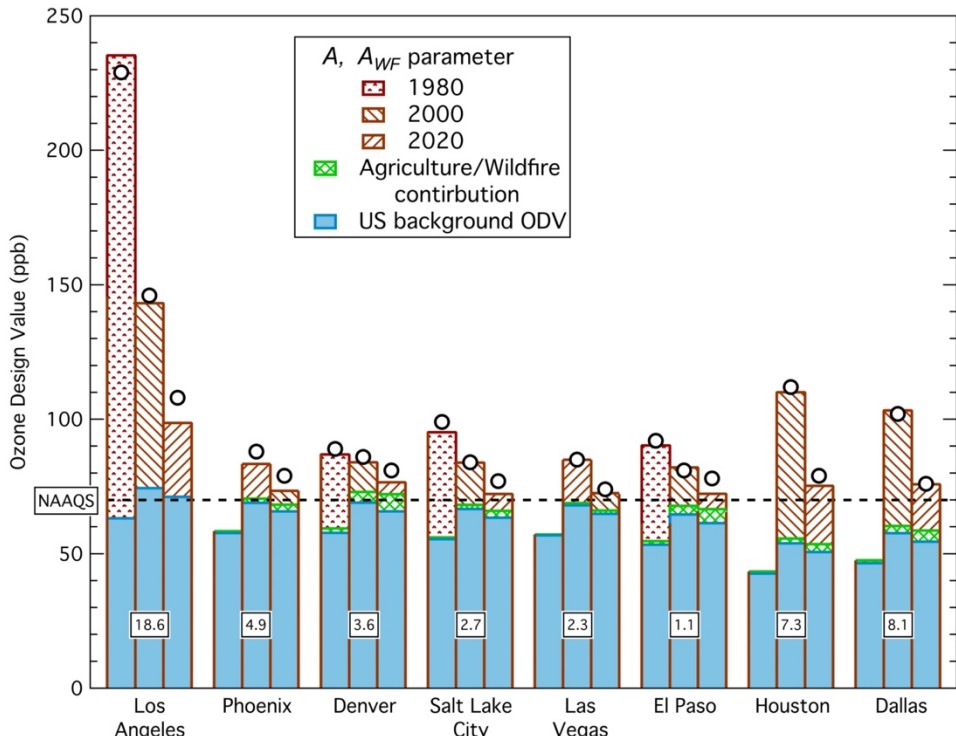

**Figure 12:** Derived apportionment of maximum ODVs for 3 years at 20-year intervals for the Crestone site in the Los Angeles urban areas (i.e., California's South Coast Air Basin, SoCAB), for four southwestern US and three Texas urban areas. The lower, solid blue bars indicate the US background ODV; they exclude estimated agriculture/wildfire contributions, which are separately indicated by the middle, cross-hatched green bar segments. The top, brown patterned bars indicate the estimated US anthropogenic ODV enhancements, again excluding the agriculture/wildfire contribution. For four cities the US anthropogenic ODV values are missing in 1980 since the tabulated ODVs are inadequate to estimate that parameter that early in the monitoring record. The 2020 populations of the urban areas are annotated in millions. The circles indicate the actual maximum ODVs recorded in the respective years in each area.





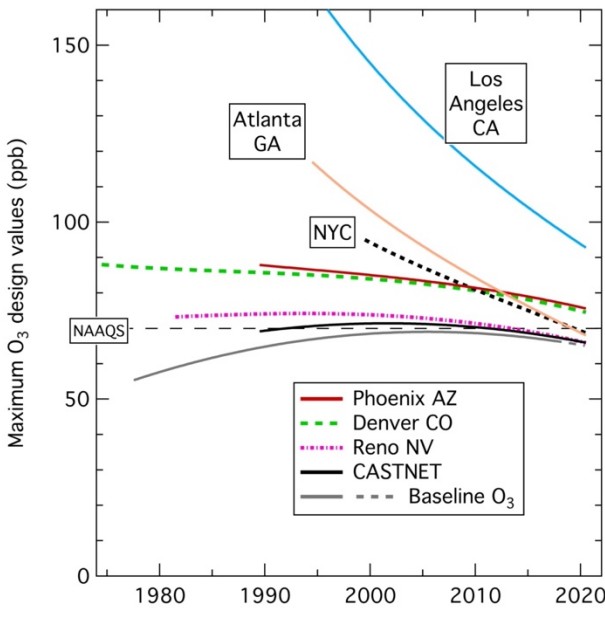

**Figure 13:** Comparison of fits of Equation 4 to time series of maximum ODVs recorded in three of the nation's largest urban areas and three southwestern urban areas, with the fits to normalized rural western CASTNET ODVs and baseline ozone from Figure 1. The fitted urban curves are taken from Figure 2 of Parrish et al. (2022) for Los Angeles, Figure S7 for New York City (i.e., NYC) and Atlanta GA, Figure 5 for Denver CO, Figure 6 for Phoenix AZ and Reno NV, and Figure 1 for CASTNET and baseline ozone.