# Peer review of "Maximum ozone concentrations in the southwestern US and Texas: Implications of growing predominance of background contribution"

_EGUsphere, 2024_

## Author Response (AR1)

**Author's Response to interactive comments by 3 Anonymous Referees on "Maximum ozone concentrations in the southwestern US and Texas: Implications of growing predominance of background contribution" by D.D. Parrish, I.C. Faloona, and R.G. Derwent**

The authors greatly appreciate the many comments made by the referees regarding our submitted manuscript; the significant time and effort that went into these reviews are clearly apparent. We have taken these comments to heart. Responses to all follow, and where appropriate the manuscript has been revised as described herein, and indicated in the "tracked changes" copy of the revised manuscript.

To organize this response, we first give an overview of the three major issues raised by the referees with a brief summary of our response to each. This is followed by point-by-point responses to all reviewer comments.

**Overview: Major Issues Raised by the Referees and Our Responses**

A. *Referee #1 questioned the appropriateness of our strong advocacy for our observation-based modeling approach and favoring it over CTMs.*

   Response and Changes Made: We retain our strong advocacy for our observation-based modeling approach, but have clarified that we definitely do not favor our model over CTMs. Rather, we now emphasize more clearly and strongly that a hierarchy of models, including both CTMs and simpler models, such as our observation-based approach, is required to fully understand tropospheric ozone.

B. *Referees #1 and #2 find that the overall organization and flow of the paper and the clarity of some of the discussion required improvement.*

   Response and Changes Made: The referees made many suggestions regarding potential improvements to the readability of the paper. We have implemented most of those suggestions; others, primarily regarding material designed to preemptively address questions that may arise with readers, such as those of Referee #3, have not been accepted.

C. *Referee #3 questioned the design and utility of our model.*

   Response and Changes Made: The referee raised several criticisms of our analysis, most of which have been previously rebutted during reviews of our earlier papers. We included discussion of some of these issues in our manuscript, to hedge against such previous contrary arguments. We have retained much of that material in the revised paper. Below we include point-by-point responses to the specific issues raised by Referee #3.

**Point-by-Point Responses**

Note: The comments from the Referees are reproduced below in black regular font with our responses in *blue italic font*.

**Anonymous Referee #1**

The authors of this manuscript present an observation-based model of ground-based ozone measurements across the U.S., including a closer analysis of the U.S. southwest region and several urban areas of interest within it. Using a regression model fit to observational data, they seek to separate U.S. anthropogenic from U.S. background ozone, and perform an analysis of trends in both ozone categories over time. Their results highlight the decline in U.S. anthropogenic ozone and the dominance of background ozone in the southwestern U.S. They also advocate strongly for this observation-based modeling approach over the use of chemical transport models (CTMs).

My general comments are as follows, with line-by-line comments below.

First, the introduction provides a critique of CTMs and speaks strongly in favor of the observation-based model that this paper will use. From the reader's perspective, this seems out of order given that we haven't yet come to understand how their model works and what it offers to justify favoring it over CTMs. I suggest relocating some of this content to the Discussion section. At the same time, it seems that there should be a place in the air quality research community for a combination of observations, observation-based models, and CTMs in the effort to understand ozone behavior and spatiotemporal trends. Rather than ranking one approach over another, they should be viewed as complementary. Moreover, ozone benefits from high spatial coverage of ground-based measurement networks that lends itself to observation-based modeling, whereas many other air pollutants do not have nearly this much measurement data upon which to judge regulatory efficacy. It seems beneficial to continue developing CTMs that could be validated against the large networks for ozone and PM that could then also be applied to more sparsely-measured pollutants. Discounting the validity and utility of CTMs seems counter-productive.

*We completely agree with the referee that there must "be a place in the air quality research community for a combination of observations, observation-based models, and CTMs." We have retained a revised version of our discussion of this issue in the Introduction. As noted in the* **Issue A** *summary above, this material is not intended as a critique of CTMs; rather it is intended as a rationale for applying our simple, observation-based model (in addition to CTMs) in a field that has for decades relied primarily on computer based modeling, rather than observational analysis. In the Introduction we have clarified our belief that "... we need a model hierarchy on which to base our understanding, ..."; this hierarchy must include both "elaborate" models (i.e., CTMs) and "elegant" models such as ours that are "only as elaborate as (they need) to be to capture the essence of a particular source of complexity, but (are) no more elaborate".*

Second, the work presented here hinges heavily on earlier work published by the same group of authors in 2021 and 2022, as well as prior works by larger groups of authors in 2017 and 2019. Could the authors comment on (a) how the present manuscript builds upon but is also distinctly different from these earlier works, …

*Thank you for this suggestion. We have moved much of the original Methods section material to the Introduction, where we develop the method description through references to prior work, rather than repeat that development in this paper. The first paragraph of the Introduction now describes how the present manuscript builds upon, but is also distinctly different from, these earlier works.*

… and (b) how their approach and outcomes are comparable with other groups pursuing the same questions in the literature? Lines 337-343 address question (a), but this comes at the start of the Results section and it may help the reader to understand these points earlier in the manuscript.

*The discussion of the need for a hierarchy of models is meant to generally address how our work complements other approaches described in the literature. We now include in the first paragraph of the Introduction a discussion of how the present work relates to our previous papers.*

Third, throughout the manuscript the authors oscillate between the terms baseline ozone, background ozone, and ozone design values (ODVs). While their analysis focuses on ODVs, they also discuss both background and baseline in relation to their own work and the work of others. Over the course of the manuscript, when jumping between terms even in the same paragraph, for a reader it gets difficult to recall which is which. The authors could handle this more carefully, as these terms are easily confused but are also related.

*We agree that these three related terms can lead to confusion. However, each of these terms has a distinct meaning, and in our discussion we must use all three terms in accord with those meanings. We have attempted to minimize possible confusion by, first, carefully defining each term as it was introduced, second, by collecting those definitions in the text box included in the introduction, and finally and most importantly, using the terms correctly and consistently throughout the paper, without any erroneous oscillation. We have carefully reviewed the definitions, text box and all discussion, and corrected all shortcomings that we found, as indicated in the track-change manuscript.*

Abstract

Line 16 refers to ODV as "an *extreme value statistic of relatively rare* maximum ozone concentrations upon which the NAAQS are based". At first read, this came across as a critique of the meaning/utility of this metric for the NAAQS. Is it meant in this way? It may be of greater utility to define it first in the way that the NAAQS do (which appears to be the same as MDA8 based on the first paragraph of the introduction) and then if the authors want to emphasize that these values are extreme and rare, this could be added after the definition of the term.

*To minimize confusion and length of the Abstract, we think it best not to define the ODV there; the ODV is related to, but not the same as, MDA8. To address this comment, we modified the discussion to first introduce ODV as the statistic upon which the NAAQS are based, and then emphasize that these values are extreme and rare. The two sentences in question now read: "We primarily analyze ozone design values (ODVs), the statistic upon which the US National Ambient Air Quality Standards (NAAQS) are based. The ODV is an extreme value statistic that quantifies the relatively rare maximum observed ozone concentrations; thus, ODV time series ….".*

Line 25 onward: The five points made here read more like findings, rather than implications (as they are named on line 25). I suggest the authors reorganize this to first clearly state the findings of their analysis, and then follow with the implications of those findings. These five points also do not need to be numbered and can just be provided as sentences incorporated into the text. Also, is point #3 only applicable to the southwestern U.S.?

*Thank you for this comment. We now clearly state three findings, and then follow with two implications of those findings. We have removed the numbering and instead use sentences incorporated into the text. Point #3 discussed preclusion of NAAQS attainment in the southwestern US; yes, it is only applicable to that region, since other regions have lower US background ODVs (see Figure 11).*

Introduction and Background

Line 57: Can the authors clarify why they included Texas with the rest of the southwestern U.S.? Is it because the urban areas in Texas violate the NAAQS? Could they also define southwestern U.S. by listing the states in parentheses?

*Thank you for this suggestion. The identified sentence has been revised and expanded to address this and other comments that requested revisions to the introduction; that discussion now reads: "Our goal in this paper is to analyze this record to quantify the sources of maximum ozone concentrations in the southwestern US, and to investigate the implications of the results. This region has not previously been analyzed by our approach, and is of particular interest because it is impacted by large background ozone concentrations (e.g., Lin et al., 2012b; Zhang et al., 2020). Previous work shows that the background contributions to ODVs in some western US regions exceeds 60 ppb (e.g., Langford et al., 2022), and even has approached 70 ppb, making achievement of the 70 ppb NAAQS quite difficult in those regions (Cooper et al., 2015). The five states - Arizona (AZ), Colorado (CO), Nevada (NV), New Mexico (NM), and Utah (UT) – included in this region have four urban areas – Phoenix AZ, Denver CO, Las Vegas NV and Salt Lake City UT– that are centers of Marginal or Moderate Ozone Nonattainment Areas (see US EPA Green Book 8-Hour Ozone (2015) Area Information, last accessed 27 January 2023); additionally Phoenix AZ and Denver CO are classified as Moderate and Severe-15, respectively, Ozone Nonattainment Areas under the 2008 ozone NAAQS (see US EPA Green Book 8-Hour Ozone (2008) Area Information, last accessed 27 January 2023).) We also include Texas in this analysis because it represents a transition region between the southwestern US and the very different Gulf of Mexico region; four Texas urban areas – Dallas, Houston, El Paso, and San Antonio – are centers of Marginal to Moderate ozone Non-attainment Areas designated under the 2015 NAAQS (US EPA Green Book 8-Hour Ozone (2015) Area Information, last accessed 31 January 2023); additionally Houston and Dallas are classified as Severe-15 Non-attainment Areas under the 2008 Ozone NAAQS (US EPA Green Book 8-Hour Ozone (2008) Area Information, last accessed 31 January 2023). We examine the ODV time series recorded in these four urban areas, plus five other Texas regions containing smaller cities as well as vast rural areas."*

Lines 70-71: Could the authors offer a very brief explanation of how Parrish et al (2017) isolated the U.S. anthropogenic ODV? This seems like an important piece of earlier work upon which the current method in this manuscript hinges.

*We have moved the development of Equations 1-3 to the introduction, so the separate quantification of the US anthropogenic ODV enhancement and US background ODV is now fully discussed in the Introduction.*

Lines 75-78: Can the authors offer a citation for this statement that baseline ozone constitutes the large majority of the U.S. background ODV? Is this also based on the Parrish et al. 2017, 2019, 2020, and/or 2022 manuscripts that are cited here?

*Parrish et al. (2022) gives the clearest demonstration of this statement. Zhang et al. (2020) is an example of a modelling study that found a similar result stating, "background $O_3$,…, is an important source of the observed year-to-year variability in high-$O_3$ events over the WUS during spring." The following sentence now begins the description of the method development, which is located immediately after the text box: "Prevailing westerly winds carry midlatitude Pacific marine air into the Western US; the baseline ozone in that air provides the predominant source of the US background ODV in the western US (Zhang et al., 2020; Parrish et al., 2022)."*

Lines 84-86: Are these the same definitions of anthropogenic and background that are used in other similar studies? It seems very notable that domestic precursor emissions from agricultural soils, livestock, and VCPs fall under "background" by the definition here and it would be helpful to know if that is consistent with or different from the broader literature.

*We cannot definitively speak to the use of the term "background" in the broader literature, but we have found that these terms are used with a wide variety of differing meanings. Our goal in the lines cited by the referee is to make clear that 1) domestic precursor emissions from agricultural soils, livestock, and VCPs must be considered anthropogenic, but 2) our simple model may possibly include ozone produced from these emissions as part of the "a" parameter, and hence appear to be "background", if we are not careful to consider these unregulated influences in our discussion. We have expanded and clarified the above cited lines to read: "Nevertheless, it is important to draw a careful distinction between the definitions of US background ODV and US anthropogenic ODV enhancement presented above, and their observation-based quantifications that we derive. Since those quantifications rely on the observed temporal changes in ODVs, only sources of precursors effectively controlled by regulatory action can contribute to the derived A parameter value, i.e. the estimated US anthropogenic ODV enhancement. The impact of uncontrolled anthropogenic precursor emissions, such as those from agricultural soils, livestock operations, or volatile consumer products, for example, would potentially contribute to the a parameter value, i.e. the estimated US background ODV. Drawing conclusions from the derived quantifications must carefully consider this issue, and bring in additional considerations to avoid confounding influences."*

Lines 88-89: If a rare, transboundary event occurs on top of typical background conditions, then it would seem that the two are in fact additive. Can the authors explain this further?

*Here we mean that the background (including both rare, transboundary events and typical background conditions) and the anthropogenic contributions are not necessarily exactly additive; we have clarified this sentence: "Furthermore, the ODVs represent rare events that may occur at any time during the warm season when the combined USB and anthropogenic ozone contributions maximize. This maximum may not occur on the days when the USB*

*contribution is largest (i.e., the days that determine the US background ODV). Thus, for any given extreme ozone episode the sum of US background ODV and the anthropogenic contribution to that episode may not equal the observed ODV; Section S2 of the Supplement discusses in detail the distinction between the US anthropogenic ODV enhancement and the actual anthropogenic contribution to an ODV."*

Line 90: Can the authors elaborate on what they mean by "when accurately determined"? As a reader this raises the question of examples where U.S. background ODVs have not been accurately determined.

*This phrase is confusing at this point; it has been removed. It referred to the importance of drawing a careful distinction between the definition of US background ODV and its observation-based quantification that was mentioned in the first sentence of this paragraph, and is now more clearly addressed by our response to the preceding comment.*

Line 94: At this point it is still unclear what "the above referenced observation-based analysis" actually is. Without having to also read Parrish et al (2022) in detail, could the authors provide a brief but more specific description of that analysis here? It seems to be a critical foundation upon which the current work builds.

*We agree that this is a critical foundation upon which the current work builds. However, we have not found it possible to give "a brief but more specific description of that analysis". Instead, we have moved the development of the observation-based analysis from the Methods section to the Introduction to serve as that foundation.*

Lines 108-144: (Related to the general comment above) This section spends a fair bit of text justifying the use of their observation-based model in favor of CTMs, but we haven't actually seen what their model is or how it performs yet. So, the detailed CTM critique and comparison feels preemptive. Much of this section actually reads like discussion or conclusions. I suggest that the authors focus the Introduction more on the underlying ozone problem they seek to understand, and available methodologies to do so, which will inform the observation-based modeling approach they are about to show us. A discussion of what CTMs can and cannot currently do is warranted, but I suggest the authors withhold judgement on the strength of their methodology over CTMs until the conclusions when they have provided evidence to support this argument.

*(Please see our response to the referee's first general comment above.)*

Data sets

Lines 158-168: Earlier it seemed that ODVs and MDA8 values are the same, but this paragraph would suggest otherwise since it is stated that MDA8 was used for the California air basins and ODVs for the rest of the U.S. Can the authors please clarify this? It is also unclear from this text whether exceptional events were included or excluded (either by choice of the authors or by reporting to the EPA database). Though it appears Section S6 elaborates more on the impact on the exceptional events, can the authors briefly state here how they determined that the analysis in this paper was not significantly affected?

*It is important to note the distinction between ODVs and MDA8 values. At a particular site, a maximum daily 8-hour average (MDA8) ozone concentration occurs every day; thus, in a year with complete data coverage, 365 (or 366) MDA8 values are recorded. The ODV statistic is defined as the 3-year running mean of the annual 4th-highest MDA8 ozone concentration at a site. Thus, the ODVs are derived from MDA8 values, but are calculated from only the very largest MDA8 values.*

*Exceptional events were included in the data set; we have clarified the text to read: "Exceptional events, such as wildfires or stratospheric ozone intrusions, are included in the data set, although in principle they can be removed from the MDA8 monitoring record through an EPA concurrence process as uncontrollable "Exceptional Events", thereby altering the ODV archive; however, the analysis presented in this paper is not significantly affected because EPA has only rarely concurred in such removal in the present study region. Section S6 of the Supplement more fully discusses this issue."*

Methods

Line 184: Is it the case that ODVs are in fact subject to diurnal and seasonal variations, given that the highest MDA8 values are almost certain to occur in summertime at midday? It is unclear how the 3-year averaging would eliminate this dependence.

*The line is correct as originally written. Since only a single ODV is tabulated for a given site in any particular year, that single value cannot convey any information regarding diurnal and seasonal variations.*

Line 215: For readers unfamiliar with Parrish et al (2022) can the authors briefly explain what this approach entails? This would aid in following the subsequent presentation of their mathematical model. Or is it the case that the method discussed on line 216 is the same as the Parrish et al (2022) approach? What elements are the same and what elements are different?

*To the first paragraph of Section 3 Methods we have added the following sentence: "The analysis utilized here is generally the same as that of Parrish et al. (2022), which is reviewed in the Introduction and Background Section 1, with the additional development of an estimate for the average, long-term contribution of wildfire emissions to urban ODVs (Section 3.2)."*

Lines 231-236: Can the authors offer a bit more explanation on *why* the a, b, and c terms represent the stated components of temporal ozone behavior? How confident are the authors in the "direct physical interpretations" of these values?

*We have removed the development of Equation 1 from the methods, so the discussion of the representation of the b and c terms has been removed. This has been replaced by related discussion: "The positive value of b and the negative value of c indicates that baseline ozone concentrations increased before 2000, reached a maximum after 2000, and then decreased at later times. Equation 2 gives the year of the maximum of the fitted curve,*

$$year_{max} = -b/2c + 2000, \qquad\qquad (2)$$

*which is ~ 2006 for the above parameter values." We are quite confident in these interpretations, since they follow from direct mathematical manipulation of Equation 1.*

Lines 244-245: Can the authors specify where the observations were generated that resulted in these values in Parrish et al (2020)? The "northern midlatitudes" are mentioned but is this from sites in the U.S.? Europe? Asia? Multiple continents? And how many sites? This would help readers to understand how applicable these values are likely to be for the sites in question in the present manuscript (since they are in fact used here).

*This description (now in Section 1) has been expanded: "… that were common to eight northern midlatitude baseline ozone time series measured from surface sites, aircraft and sondes over western Europe and western North America covering altitudes from sea level to 9 km."*

Line 252: Is this statement about urban ODVs decreasing rapidly and approaching non-zero background based on results from this work or from an earlier publication? If it is the latter, then could a citation please be provided? This is important for readers to understand how the A*exp(-t/T) term was derived.

*This statement has been replaced by more extensive discussion in Section 1, which is fully referenced.*

Later in the paragraph they say that more information on the choice of the exponential function is described in the supplement but a brief explanation would be helpful here.

*The penultimate sentence in that paragraph (now in Section 1) has been modified to read: "An exponential function is chosen to quantify the long-term decrease of US anthropogenic ODV enhancements, because it is mathematically as simple as possible (i.e., has the fewest possible unknown parameters), and successfully accounts for a large fraction of the variance in recorded ODV time series throughout the US (see Parrish et al., 2017; 2022; Parrish and Ennis, 2019); this choice is more fully discussed in Sections S3 and S4 of the Supplement."*

Line 257: Earlier the authors said that the value for c would be derived from the present work (Line 237) so how should readers interpret the use of a previously derived value for c here?

*Thank you for this comment as our discussion was confusing. Since we have moved the development of Equation 3 to Section 1, this confusion has been removed. We have added to Section 3.1 further explanation for the two values of c: "We also fit Equation 1 to the ODVs recorded at some isolated rural sites in the western US (Section 4.1) to verify that the b and c parameter values quantified by Parrish et al. (2020) are appropriate for general application to southwestern US ODVs."*

Line 279: Does the Iglesias et al (2022) citation apply only to the third point in this sentence, or to all three points? The authors have already cited the works by Westerling in support of point #2, but can at least one citation be provided in support of point #1?

*The Iglesias et al. (2022) citation has been removed from this line. Two citations have been added in support of point #1: (McKeen et al., 2002; Parrish et al., 2022).*

Line 285: Where is the factor of 4 represented in the model?

*From 1980 to 2020, t increases from -20 to +20 and the factor (1 + 0.03\*t) in Equation 4 increases from (1 - 0.6 = 0.4) to (1 + 0.6 = 1.6), which amounts to a factor of (1.6/0.4 = 4) increase.*

And how do the authors know the location specific ODV enhancement due to wildfires (the WF parameter)?

*The first 2 sentences of the 2nd paragraph of Section 3.2 (previously Section 3.3) have been revised to more fully explain the analysis: "In principle, fitting Equation 4 to an ODV time series would allow determination of values for three parameters (a, $A_{WF}$ and WF, with $\tau$ already fixed), but in practice such fits generally do not allow precise determination of all three parameters. However, if a can be determined from a separate analysis, then precise values can be determined for $A_{WF}$ and WF from fits of Equation 4."*

Line 288: Similar to the previous question, how do the authors know which locations have significant enhancements of ODVs due to wildfire emissions? Are these factors predicted by the data or informed by some a priori knowledge?

*The third sentence of the final paragraph of Section 3.2 has been revised to explain which locations have significant enhancements of ODVs due to wildfire emissions: "We follow this approach in the analysis of time series of maximum ODVs recorded in southwestern US urban areas, which are the only areas in the region with large enough local $NO_X$ emissions to cause significant enhancements of ODVs due to wildfire emissions."*

Lines 302-309: This in part addresses an earlier comment about the exclusion of these precursor emissions from the definition of U.S. anthropogenic ozone. It would be helpful to circle back on this point for specific regions or urban environments in the discussion of results in this manuscript. It seems like this exclusion would have more notable impacts on some urban areas (e.g. Denver) more than others, and it would help to acknowledge this based on the body of literature that has explored some of those locations specifically.

*The following phrase has been added to the 3rd to last sentence in Section 3.2: "...:the Front Range of Colorado, including Denver (e.g., McDuffie et al., 2016) and Dallas-Fort Worth (e.g., Ahmadi and John, 2015) are urban areas where such impacts have been subject of several studies.*

Results

Lines 332-336: This paragraph would fit well at the end of the introduction or start of methods, so that the objective of the methods is easier for readers to understand. (The end of line 334 also appears to contain a typo).

*We intend this short paragraph to be a preview for the reader of what is covered in the Results section. The final paragraph of the Introduction section already discussed this information in more general terms. (And thank you, the typo has been corrected.)*

Line 346: Please also state for the reader what sections 4.3 and 4.4 will analyze.

*Lines 345 and 346 have been changed to read: "… in the CA coastal air basins, and the following three sections present the analysis of ODV time series in the southwestern US (Section 4.3) Texas urban and rural areas (Section 4.4), as well as some in other US states to provide context (Section 4.5).*

Line 370: Earlier it was stated that coefficient b was derived from earlier work and applied here, whereas coefficient c would be derived here (lines 244-245). But this sentence implies different b coefficients by site. Can the authors clarify this?

*Thank you for this comment: Lines 244-245 stated that "Parrish et al. (2020) derived numerical values for the second and third coefficients in Equation 1 - $b = 0.20 \pm 0.06$ ppb $yr^{-1}$ and $c = -0.018 \pm 0.006$ ppb $yr^{-2}$ - …. We use these same coefficient values in this work." This is correct for most of the analysis. However, in Section 4.1 we derive values of b and c coefficients from Equation 1 from ODVs recorded at the remote rural CASTNET sites. We have clarified this issue by retaining the original wording of lines 244-245 (but now moved to Section 1) and added the following sentence to the 1$^{st}$ paragraph of Section 4.1: "Here, we first fit Equation 1 to ODVs from such sites to derive b and c parameter values specific to the western US ODVs and compare those values to the values cited above from Parrish et al. (2020), and second, fit Equation 3 to these same time series to determine the small US anthropogenic ODV enhancement at those sites."*

Line 415: change "caused" to "cause"

*Thank you; corrected.*

Figure S1 is referenced frequently throughout section 4.2, and is the subject of the first sentence of the discussion online 569. I suggest the authors consider moving this figure to the main body of the text if it is going to be a focal point of discussion and not simply supporting information. If the current Figure 3 is sufficient to make their points, then the Results and Discussion should focus on this figure, with a direction to the supplement for more detail. But in the current format it seems that the reader needs to see all of Figure S1 to understand the discussed outcomes.

*Thank you for this suggestion. The original Figures S1 and 3 have been combined into a new Figure 3, with Figure S1 eliminated.*

Line 453: Could the number of rural sites considered be added in parentheses?

*Line 452 modified to read: "Monitoring sites also sparsely cover the rural areas in these five states (19 in CO, 9 in NM, 6 in UT, 8 in AZ and 2 in NV as shown on map in Figure S1)."*

Line 528: There appears to be a typo at the end of this sentence: "in southern and eastern Texas in the Southwest Texas"

*Thank you. This phrase has been corrected to: "in southern and eastern Texas in the SW Texas region". The label of that region in the upper right graph of Figure 8 has also been changed to "SW Texas".*

Section 4.5 reads as supplementary information, and could be reduced to a couple of sentences for the main body of the paper.

*This Section analyzes ODVs from regions of the country that have not been previously discussed in our other publications. We believe that it is important to include that short discussion in this section.*

Discussion and Conclusions

(see comment above about moving Figure S1 to the main text)

*(Please see our response to that comment.)*

Lines 573-574: How do the authors know that 0.6-1.0 km is "high enough to avoid continental influences as air is transported ashore and low enough to represent air mixed into the convective boundary layer over the continent". Is this a novel suggestion from these authors or supported by the literature (for which citation(s) could be provided)?

*The sentence on these lines has been modified to read: "The ozone altitude gradient (see Figure 1 of Parrish et al., 2022) indicates that this altitude range includes the MBL to free troposphere transition, so that it is high enough to avoid most continental influences as air is transported ashore and low enough to well-characterize air mixed into the convective boundary layer over the continent (Parrish et al., 2010).*

Lined 596-606: These sentences provide very helpful introductory material to set up the motivation for trying an observation-based model and highlighting the challenges or missing elements of CTMs. I suggest to move these sentences to the Introduction, and in exchange move comments from the Introduction that compare the observation-based model to CTMs to some part of Section 5, now that we can see how the authors' model performs.

*We have chosen to retain these lines in the Discussion and Conclusions Section. Since we are not attempting to critique CTMs in favor of our observation-based model, we believe that the results of our analysis indicate one reason that CTMs have shortcomings, so this implication of our findings does belong in this final section.*

Much of Section 5.2 also reads as introductory background material, and very minimally integrates discussion of the new results obtained in this work (including Figure 9, which is mentioned here but as far as I can tell is not derived from the model they developed for this manuscript). I think this section would actually set the stage nicely for why the authors are motivated to do this observation-based model analysis, and suggest it be integrated into the introduction as helpful framing for the reader.

*Here we disagree with the Referee. In our view, a scientific paper should investigate further issues that are directly implied by the analysis in that paper; this is the goal of section 5.2. Our analysis led us to conclude that it has been and will continue to be very difficult (if not impossible) to attain the ozone NAAQS in the southwestern US. One implication of this conclusion is that this difficulty should be reflected in the history of NAAQS exceedances. The referee is correct that this analysis is in addition to the model we developed for this manuscript,*

*but as far as we are aware the analysis from the same or a similar approach to ours is novel in this paper. By investigating this implication of our original analysis, we provide direct support for the utility of our original model and its results. We believe that this material definitely belongs in the Discussion and Conclusions Section, not in the introductory background material.*

Line 668: Have other CTM analyses been published more recently with higher resolution models, as model development has continued to improve? Or is 200x200km the best available right now?

*CTM analyses at higher resolution have been published. For example, the GFDL-AM4 model in the Zhang et al. (2020) reference that we discuss is run at ~ 50 x 50 $km^2$ horizontal resolution. However, the Jaffe et al. (2018) reference from which Figure 10 is taken is the reference often cited for policy relevance, which dictated our choice.*

Line 687: Text refers to the "Crestline" site but Figure 12 caption refers to "Crestone". One of these appears to be a typo?

*Thank you for identifying this typo. "Crestline" is correct; Figure 12 caption has been corrected.*

Line 716 (or elsewhere): what changes in ozone were observed in the urban areas examined here during the initial COVID-19 lockdowns? In the absence of local transportation emissions, did urban NO2 and ozone decrease in any of these cities? This information may not be readily available for all cities under consideration, but would be an interesting piece of evidence to consider with regards to the magnitude of the US anthropogenic contribution. It may, however, also be the case that the lockdowns were not long enough and not during peak ozone season for such information to be impactful. But it may be worth investigating.

*This issue has been investigated from an observational perspective. The final paragraph of Section 2 of our manuscript summarizes our thoughts: "The ozone data considered here include the time period of emissions reductions resulting from societal efforts to control the COVID-19 pandemic. Many publications have examined the impacts of those emission reductions on ozone in areas throughout the world, with widely varying findings. No consistent impact has been found for summertime maximum ozone concentrations (e.g., Gkatzelis et al., 2021), which are the focus of this study. We find no apparent systematic deviations in MDA8 concentrations or ODVs reported for 2020 – the year of largest emission reductions – in any of the time series examined, so this issue is not considered further in this analysis."*

*This issue is also the subject of model simulations. A recent publication (He et al., 2024) concludes: "Over a full year (April 2020 to March 2021), COVID-induced emission reductions led to 3–4% decreases in national population-weighted annual fourth maximum of daily maximum 8-h average $O_3$ and annual $PM_{2.5}$." That ozone decrease corresponds to a decrease of ~ 1.4 ppb, and they note noticeably larger decreases over Southern California ($-1.7 \pm 0.8$ ppb). In contrast, an earlier modeling study (Jiang et al., 2021) showed emission reductions led to small increases in MDA8 $O_3$ over urban areas in Southern California during the lockdown period.*

*In view of these references, we judge that we are fully justified in not considering this issue further.*

Line 734: The comparison of ozone to ionizing radiation is an interesting one. But is it not a bit of an apples-oranges comparison when you consider how ozone is transported, photochemically produced, and thus a secondary transboundary air pollutant that can cross state lines? Where do you draw the boundary for a southwestern U.S. standard?

*We agree with the Referee's comment. We have added a cautionary sentence to the paragraph discussing this issue: "Of course, such an approach would bring additional complexity, since ozone is secondary air pollutant that can cross state lines, making it difficult to define a suitable boundary for any particular region, such as the southwestern US."*

Line 746: this recommendation to offset wildfire ozone by lowering urban NOx seems like it would depend on whether the urban environment was NOx or VOC limited. Can the authors comment on this for Denver or any other urban area?

*Since wildfire plumes are VOC-rich and NOx-poor, in areas (urban or other) where wildfires significantly increase ODVs we can conclude that those areas are VOC limited, at least during the days when the $4^{th}$ highest MDA8 ozone is recorded. We cannot make a more general conclusion.*

*We have slightly modified that material to read: "..., , although in addition the small wildfire contributions possibly can also be reduced indirectly through decreased $NO_X$ emissions in VOC-poor urban areas."*

Also in terms of Denver, the definition of U.S. anthropogenic in this manuscript does not include agriculture or oil and gas – this is missing a key source of local ozone in the Denver Basin that is well-documented in the literature and does not seem to fit the category of "background" given that local efforts could in theory reduce the contribution of these sources to Denver's ozone production. Can the authors please comment?

*Here is an area that we are not in total agreement with the Referee. We do agree that emissions from both agriculture (primarily animal husbandry) and oil and gas development/production are important in the Colorado urban Front Range, and that they play some role in local ozone production. However, we are not convinced that they significantly elevate ODVs; in our opinion the published literature has not demonstrated such an effect. However, we do agree that this is an open scientific question.*

*To address this comment, we have added the following as a $4^{th}$ paragraph to Section 4.3: "An open scientific question is the role of emissions from the oil and natural gas industry and agricultural activities in elevating ODVs in the Denver urban area. To the extent that emissions from these sectors have increased similarly to wildfires, any such role would contribute to our derived WF parameter value, which is larger in the Denver urban area than other urban areas considered here (see Table S4). With a different temporal dependence they could bias the derived a parameter value; however, the good agreement between that estimate in the Denver urban area, rural CO and the CASTNET data indicates no more than a small bias."*

Lines 765-771: Can the authors also comment on how (if at all) their interpretation would change if the ozone NAAQS were further lowered from 70 ppb to, say, 65 ppb? Will areas other than the southwestern U.S. still be required to reduce their local anthropogenic precursor emissions to

remain in attainment? It may be difficult to speculate on something that hasn't happened at this point, but a standard lower than 70 ppb was considered in the past and we just saw a tightening of the PM2.5 primary standard which has analogous regional effects on the challenges of emission controls. Is there a related point to be made for ozone, should something similar happen in the future?

*To address this comment the following sentence has been added to the first paragraph of Section 5.3: "Were the NAAQS to be lowered further, for example, to 65 ppb as has been considered in the past (e.g. Regulatory Impact Analysis of the Final Revisions to the National Ambient Air Quality Standards for Ground-Level Ozone), Figure 10 indicates that NAAQS achievement would be precluded over the vast southwestern US region inside the 64 ppb contour (unless regulatory efforts could reduce the US background ODV)."*

**Anonymous Referee #2**

Summary:

Background ozone is an important consideration for meeting national air quality standards and future policy. A key finding of this work is that background ozone is a major contributor to observed ozone levels in the southwestern US and that anthropogenic enhancements are small relative to this background ozone contribution. This paper brings value and insight into how we quantify background ozone in the southwestern US, how we think about anthropogenic enhancements on top of that background in this region, and what the implications are for continued emissions reductions in this region in the broader context of national air quality standards.

*Thank you; we greatly appreciate these supportive statements.*

Comments

L81: The text box of key terms is a helpful visual aid in the intro. However, the current definitions in this table are confusing. For example, US background ozone is defined as "USB" in the text-box, but that definition is never used again.

*Thank you for this observation. In the literature, there are several terms that are used with varying meanings. We believe that it is essential to clearly define and consistently use a single set of definitions throughout the paper; hence, the use of the text box to collect those definitions, which are defined in the text. We include "USB" because it is often found in the literature, we now use it extensively in our revised manuscript. We also extensively use the related term **US background ODV**, which follows "USB" in the text box in order to emphasize that relationship.*

Further, "ODV" is used to define two different things in the text box. It would help to re-define or denote the terms in a way that better assists the reader in recognizing the differences right away. For example, background ODV could be denoted as ODV subscript BKG, reported ODVs could be denoted as ODV subscript reported, and anthropogenic enhancements could be denoted as ODV subscript enh.

*Thank you for this suggestion. However, we have used a consistent set of terms that are consistently defined in three previous papers (Parrish et al.; 2017; Parrish and Ennis, 2019;and Parrish et al., 2022). We believe that changing terms, notation or definitions in this paper would add to the confusion, rather than clarify.*

Even with this table of terms, it was occasionally hard to follow when the authors are referring to ODVs reported, background ODV, or the anthropogenic enhancement in ODVs above background. So, it could also help to have more consistency in the usage and definitions of these terms.

*We agree that consistency is essential. We have carefully re-read the paper, to ensure that no inconsistency remains in our use of any of these terms.*

It was somewhat confusing to find the fit parameters in this initial table in the intro. Breaking up this information and moving the descriptions of the fit parameters to a separate text box in the methods section where the equations are discussed might be a better way of presenting this content. Adding the equations to this text box with brief descriptions of the utility each could also help serve as a quick reference guide to refer back to when reading this paper.

*Thank you for this suggestion. However, from previous experience we find it very important to include the fit parameters in the text box along with the terms for which they provide estimates. For example, the parameter **a** is simply a parameter in an equation, and fitting that equation to a time series of observed quantities provides a value for that parameter. Our model is set up so that the parameter value provides an estimate of the **US anthropogenic ODV** in year 2000. However, **a** is not the **US anthropogenic ODV**. We have found that including these short tutorial phrases in the text box provides some needed clarity in the later discussion. To avoid potential confusion, we have moved the equations to immediately below the text box, so that it is now easy to locate them within the text; thus, we do not think that a second text box is needed.*

L94-144: This section seemed like it would be better served in the discussion in section 5. Many of the same concepts are addressed in more detail in Section 5, and it seems like a better flow and usage of space to simply incorporate this content into that section. This also allows for the novel part of this work to be established sooner than L145.

*We have retained the material in these lines in the Introduction, as we believe that it is important to demonstrate the need for our work at the outset. In our revised manuscript, we now establish the goals and novel aspects of our work in the first paragraph of the Introduction.*

In general, it was my impression that the overall flow of the paper and ease of readership could be improved with some modest rearrangement. For example, the opening sentences of Sections 4.3 (L446-451) and 4.4 (L502-507) seem better placed in the introduction for context about ozone issues in the Southwestern US.

*Thank you. These sentences have been moved from Sections 4.3 and 4.4 to the introduction.*

It also seems like some sections of the main paper could have gone in the SI, while some sections of the SI should have gone in the main paper (specifically key points about uncertainties and the consideration of exceptional events).

*Here we disagree with the referee. We believe that the importance of all material in the main paper justifies its retention there. We also judge that the uncertainties of our analysis are suitably considered in the paper, with the more extensive material available in the Supplement. As we discuss in response to this referee's comment below on Section S6 and comment 4 by Referee #3, the consideration of exceptional events has at most only very limited impact on our analysis; thus, its inclusion only in the Supplement is also justified.*

L159: It would help to establish early on in this section that the study period is limited to summertime (i.e., May through September). This could be easily added to the first sentence of this paragraph.

*It is important to note that it is not correct to state that the entire study period is limited to summertime. The ODV statistic is defined as the 3-year running mean of the annual 4th-highest MDA8 ozone concentration at a site, regardless of the season in which the 4th-highest MDA8 occurred. Thus, the ODV analysis includes all seasons, although the majority of the highest MDA8 values certainly occur in summer. The analysis of the MDA8 ozone concentrations in the four coastal California air basins is limited to May through September, but we do not think that it is helpful to mention that in the description of the analyzed data sets.*

L167: If I have this correct from this section (and Section S6), the authors choose not to select the "Exclude exceptional events data" option that excludes all flagged exceptional events regardless of concurrence. Thus, all reported exceptional events in the EPA AQS datasets regardless of whether EE are reported by the state agency that monitors ozone in a given area are included in all ODV calculations in this work. It seems appropriate to include exceptional events (even if each agency's treatments of EE at individual sites may vary), since excluding them could lead to a significant difference in the annual ODV's determined for each location.

*Yes, we choose not to select the "Exclude exceptional events data" option, and we agree that this is the appropriate choice for our analysis.*

However, the authors make a subsequent statement in Section S6 that says "the analysis of this paper is not significantly affected by consideration of exceptional events", which leaves me still questioning: 1) whether the authors did or did not actually include exceptional events, and 2) how they know that it does or does not affect the results and conclusions. Can you comment on how different your results would be with and without selecting the "Exclude exceptional events data" option? An observation-based sensitivity test of this, for say a few selected urban and rural locations, could be an interesting addition to this work. This seems like an important consideration since exceptional events have been found to greatly impact the southwestern US (See David et al., 2021, https://iopscience.iop.org/article/10.1088/1748-9326/abe1f3) and ozone concentrations can be substantially elevated on smoke impacted days (e.g., Buysse et al., 2019, https://doi.org/10.1021/acs.est.9b05241).

*Again, we did include exceptional events in our analysis. Our comment in Section S6 actually reads: "In summary, archived ODVs can be reduced by US EPA exceptional event concurrences; however, to date concurrences have been extremely limited, and therefore have not significantly affected the analysis presented in this paper." We have not attempted to*

*investigate the influence of excluding exceptional events as reported by state agencies, and do not wish to undertake such an effort.*

L172: Can you add a reference for this sentence about sensitivity tests?

*These sensitivity tests were completed in response to a review of Parrish et al. (2021). Unfortunately, this response is not publicly available, so we cannot add a reference.*

L182: The phrase "during the entire periods of ozone monitoring" is a bit confusing. This seems to imply year-round measurements, yet this analysis focuses on summertime.

*As noted above in our response to this referee's L159 comment, our analysis of ODVs is not limited to summertime. No change has been made to this wording.*

L190: The authors mention that the single largest MDA8 O3 concentration is considered. Do you know if the day that this highest concentration was measured was impacted by an exceptional event? Would it change your results and conclusions if it is?

*None of the single largest MDA8 concentrations considered in the four California air basins were impacted by an exceptional event to which EPA concurred.*

L209: The authors imply that some years of data are excluded for select sites because they "do not appear to accurately represent the overall urban area". Is this fair? What criteria is used to justify excluding this data?

*We have not formulated specific, objective criteria to exclude early, unrepresentative data. We have added additional discussion of this issue, and moved the entire discussion to the first paragraph of the original Section 3.4 (now Section 3.3): "Networks of ozone monitoring sites have operated in all of the largest US urban areas; the ODVs from several of these networks are examined in detail. Where possible, the full time periods of the data records are included in the analyses. However, the analysis based on Equations 3 and 4 can only represent the time period over which ODVs consistently decreased. In some urban areas consistent decreases did not begin until after measurements began, or the early data do not appear to accurately represent the overall urban area; in this case the ODVs selected for analysis begin at a later year. The most extreme example of the latter case is Las Vegas, where several newly established measurement sites began reporting ODVs in 2000; these values were significantly larger than reported from any other sites in that urban area. Thus, the analysis of the Las Vegas data is limited to 2000-2021 to avoid a discontinuity in the time series of ODVs. In the following section, the figures that illustrate the analyses include all recorded ODVs, so selection of ODVs for analysis can be examined."*

L241: This seems like a good place to follow up with a note about not being able to forecast ODVs.

*While reorganizing the text to move the discussion of the methods to Section 1, we eliminated this sentence altogether.*

L294: There are ways to empirically estimate smoke impacted days at individual monitoring sites (e.g., Brey & Fischer, 2016, https://pubs.acs.org/doi/10.1021/acs.est.5b05218; Buysse et al., 2019, https://doi.org/10.1021/acs.est.9b05241; McClure et al., 2018, https://doi.org/10.1016/j.atmosenv.2018.09.021).

*We agree that there are ways to empirically estimate smoke impacted days at individual monitoring sites; however, none of these methods are capable of providing "knowledge of the spatial and temporal variability of wildfire impacts on maximum ozone concentrations at individual sampling sites." Please note that we have added "on maximum ozone concentrations" to this sentence.*

L310: Could oil and gas be contributing to this term? Prior studies show substantial increases in oil and gas development activities in parts of Texas and that these development activities influence much of the southwestern US (e.g., Koss et al., 2017, https://amt.copernicus.org/articles/10/2941/2017/; Dix et al., 2019, https://agupubs.onlinelibrary.wiley.com/doi/full/10.1029/2019GL085866; Pan et al., 2023, https://doi.org/10.1080/10962247.2023.2266696).

*As we note in the paragraph beginning on Line 314 of the manuscript: "There are additional anthropogenic emission sectors that may not have decreased over time, and hence could possibly bias our estimate of US background ODVs. These sources include emissions associated with oil and gas (O&G) exploration, drilling and production, which have increased over the past two decades in some regions of the Western US." and continue to note: "The Supplement Section S5 of Parrish et al. (2022) analyzes time series of ozone observations in the Bakken O&G basin located in North Dakota …. That discussion found no indications of a significant bias arising from these emissions sectors." Thus, we think that this comment has already been adequately addressed.*

L320 (and 485): The range of RMSD seems rather large compared to the anthropogenic enhancements (6 ppb on average). How does this impact the findings? Can additional bounds be put on background and anthropogenic contributions by propagating the errors associated with the variability in fits?

*As discussed in the third introductory paragraph of Section 3 Methods (now third paragraph in Section 3.1), 95% confidence limits are quantified for all parameter values derived from the functional fits. These are the bounds that can be put on the background and anthropogenic contributions, and they are specified in all figure annotations, tables and in the discussion throughout the paper. All findings are stated so as to be consistent with these bounds.*

L337 – 346 and parts of Section 4.1 seem like they could be better placed in the methods section and relabeled as "Section 3.5: Justifying assumptions for determining long-term ODV at rural sites".

*Lines 338-346 were duplicative of material in Section 3.1 that was added in response to the Line 257 comment of Referee #1, so they have been removed from this introduction to Section 4. Since many readers will begin reading the results section without first carefully reading the Methods*

*Section, we believe it is important to include this explanatory information in Section 4.1, so that material has been retained there.*

L373: The reason for normalizing the data in Figure 1 was not clear. Also, does the selection of what the data are normalized to (i.e., 71.4 ppb) matter? A sensitivity to this selection seemed to be indicated in L388, but again it was not clear if and how this matters. Does this selection bias the background ODVs derived in other fits using a similar normalized *a* parameter, as mentioned on L394?

*The introductory sentence on lines 373-374 has been clarified: "The similarity of the temporal evolution of these rural ODVs suggests normalizing the ODVs to remove the spatial gradient and to derive a single fit of Equation 1 to all ODVs from the eight sites; the greater number of ODVs in a single fit then allows more precision in the determination of the derived parameter values." The selection of the value to which the data are normalized (i.e., 71.4 ppb) does not affect the derived **b** and **c** parameter values, but does definitely affect the derived **a** value. Line 394 refers to other illustrative curves, not to other fits. There is no bias introduced into any derived background ODVs.*

L 415: Change "caused" to "cause"

*Thank you; corrected.*

L558: Does anomalously high ODVs in the Upper Green River Basin during winter impact the results of this summertime study? Is it fair to exclude summertime data from this site in this analysis? It also seems worth mentioning that this area is also notoriously impacted by oil and gas production activities (e.g., Ghimire et al., 2023, https://doi.org/10.5194/acp-23-9413-2023).

*As noted above in our response to this referee's L159 comment, our analysis of ODVs is not limited to summertime. It is not possible to keep the summertime data for Boulder Wyoming site, since we analyze the EPA-tabulated ODVs, which are primarily affected by the wintertime data at this site. Since we base our interpretation of the anthropogenic enhancement of ODVs on photochemical production from other anthropogenic precursors, which does occur primarily in summer, we do believe that it is important to exclude the ODVs from this one site. We have added a mention of the oil and gas production activity in this region; the phrase on lines 557-558 has been modified to read: "...; since this site is in the Upper Green River Basin where high ozone concentrations are produced in wintertime from the region's large oil and gas production activities (Schnell et al., 2009), all ODVs from this site have been excluded."*

L643: Could recirculation of air associated with the "Denver Cyclone" (Reddy and Pfister, 2016, https://doi.org/10.1002/2015JD023840) lead to an overestimate of background and/or anthropogenic ODVs in parts of Colorado?

*We do not expect that the "Denver Cyclone", or any other meteorological recirculation pattern in any region, can affect our analysis of the background and/or anthropogenic ODVs because, while they may enhance high ozone events that impact the ODVs, those enhancements will still respond to emissions reductions. In other words, the long-term temporal behavior of the two ODV components are expected to be the same, regardless of the site specific meteorological circulation patterns. To the extent that these patterns change under a warming climate, the long-*

*term progression of air quality may also change; however, we judge that this second order issue is well beyond the scope of our current analysis. In fact this may be a case where the judicious use of CTMs may be better suited to answer such questions.*

L 761: Change "his" to "this"

*Thank you; typo corrected.*

L 765: It is worth noting that reducing O3 precursors in general has more benefits related to air quality and climate than what has been put forth here. For example, reducing emissions from sources that generate VOC and NOx have the added benefits of reducing other criteria air pollutants and air toxics, co-emitted greenhouse gases, and precursors to fine particle formation and SOA which contribute to visibility and haze. While this is implied in Zhang et al. 2019, it could be more explicit stated in the concluding remarks.

*This is an excellent point, and one we have made in another manuscript (Derwent et al., 2023). The following sentence has been added to the end of this final paragraph; "Moreover, control measures that reduce VOC and NOx emissions have the added benefits of reducing other criteria air pollutants, air toxics, co-emitted greenhouse gases, and precursors to fine particle formation, which contribute to haze and visibility reduction; further air quality improvement is desirable from multiple perspectives."*

L783: For completeness, support for the third author's effort seems like it should also be included in the acknowledgements and legal notice section.

*Alas, the third author had no support; his effort was pro bono. This is now explicitly stated.*

Figure 5, 6, and 8. It isn't always clear which fit is which in the figures. It could help to add a legend and/or refer to the fits by name (or their description) rather than by equation number.

*Thank you for this observation. To clarify, a legend has been added to the right graph in Figure 5, and more description is added to the captions of Figures 6 and 8.*

Also, on several occasions throughout Section 4.2 and Section S1, the authors mention "excellent fits" of the data to equation 3. However, there are cases in these figures where the fits don't appear to represent the data that well (e.g., the green dashed line and the red curve on the left side of Figure 5 and in some of the panels in Figure 6). Can you clarify what the quantitative benchmark is for an "excellent fit"?

*Thank you for this comment. We have removed most mentions of "excellent fits". One mention remains in the first sentence of Section 4.2, where we have added an explanatory phrase: "Previous analyses have shown that Equation 3 (or a closely related equation with a constant background term, rather than the small changes quantified by the $2^{nd}$ and $3^{rd}$ terms of Equation 3) gives excellent fits to ODV time series (i.e., fits that capture a large fraction of the ODV variance and/or with relatively small RMSDs), not only in urban areas (Parrish et al., 2017, 2022, Parrish and Ennis, 2019), but also in relatively isolated northern rural states (Parrish et al., 2022) and at western US rural CASTNET (preceding section) and coastal (Parrish et al., 2022) sites." In each case, the cited references give more quantitative benchmark measures.*

Further, the red fits for many of the geographical areas shown in Figures 5 and 6 and in the top panels of Figure 8 exhibit a consistent downward trend even though there is a clear uptick in ODVs in recent years. To clarify, is this because data from recent years is not included in the fit, or is this because the fit is poor for these data points. Since one novel aspect of this work is to update the trends in the southwestern US to recent years, it seems like these data points should be included in the fits. Maybe I missed something, but it seems unfair to exclude them from the fit just because they do not fit the functional form. Given the physical reasoning for this uptick (wildfires, maybe increased impacts from oil and gas), should additional terms in the polynomial be considered?

*We agree that there is a clear uptick in the maximum ODVs in recent years in several of the urban areas (6 of 13) in Figures 5, 6 and 8, which the red curves showing the fits to Equation 4 do not capture. Importantly, the data from recent years through 2021 are included in all of these fits; no recent data are excluded. The time dependence included in Equation 4 (i.e., the exponential decrease and the wildfire increase) is of too coarse resolution to capture such upticks; notably in all of these graphs, similarly-sized upticks (and downticks) have been observed in earlier years (see for example Phoenix in the mid-90's and 2010, and Salt Lake City in the mid-90's and ~2005.) Further analysis based on additional terms in Equation 4 may be a fruitful avenue for future research, as there is likely a semi-decadal mode of variability as there are in so many climate indices. Further analysis based on additional terms in Equation 4 may be a fruitful avenue for future research, as there is likely a semi-decadal mode of variability as there are in so many climate indices.*

Figure 12. Are some of the bars out of order (e.g., forward hash for 2020 comes before backward hash for 2000 for Phoenix and Las Vegas)?

*We are grateful for your sharp eye! Yes, there were errors in the figure, which have been corrected.*

Section S3: Using your illustrative example in this section, can you estimate how much ozone precursors would have to decrease and over what time frame to eliminate the less than 6 ppbv on average anthropogenic enhancement above background? This could add context to the statement in the main paper at L719.

*It is really not possible to make this estimate, since our illustrative example is based on approximating the anthropogenic enhancement by a term that decreases exponentially – such an exponential function never reaches zero, which would be required to eliminate that enhancement.*

Section S6: It is hard to believe that there was only one ozone exceptional event concurrence when there are so many reports in the literature demonstrating exceptional events associated with wildfires in the western US over the time frame of this study. For context, I'm thinking of Figures 1 and 2 in David et al., 2021 (https://doi.org/10.1088/1748-9326/abe1f3), where they show wildland fires have a significant impact on the number of exceedance days for ozone in the western US between 2000 and 2017 (specifically EPA regions 8,9, and 10). Other reports also show a substantial number of high ozone days (as many as half) at selected sites in Colorado and California are associated with wildfire smoke impacted days in more recent years between 2016

and 2022 (some examples include, but are not limited to: https://doi.org/10.1029/2022JD037700; https://doi.org/10.1029/2021JD035221; https://doi.org/10.5194/acp-22-9681-2022).

*Thank you for this comment. We have downloaded the data archive posted by David et al., 2021. They do indeed list more concurrences, but those were under a previous exceptional event rule (i.e., prior to 2016). According to their data archive, during the 2000-2017 period, nationwide 803 total days exceeded the ozone standards and were also flagged for an exceptional event in the AQS database. The majority of these (457 or 57%) were in the 5 southwestern states, CA or TX. Of these ~10% (47) were submitted to EPA for possible concurrence; these efforts succeeded for 16 (or 34%) of the 47 days. These days comprised 6 separate exceptional events: 19 July - 3 August, 2000 in UT, 3 days in June and July, 2008 in CA, 26 August 2011 in TX, 8 June 15 in UT, 18-21 July 2015 in NV, and 2, 4 September 2017 in CO. We have revised Section S6 to reflect the above discussion. Overall, these very limited exceptional event concurrences support our concluding sentence in that section: "In summary, archived ODVs can be reduced by US EPA exceptional event concurrences; however, to date concurrences have been extremely limited, and therefore have not significantly affected the analysis presented in this paper."*

In general, it would be worth re-checking the manuscript for definitions of abbreviations and acronyms. In some cases, definitions are overly abundant and in other cases they are lacking. For example, starting on Line 354, definitions for National Park (NP), National Monument (NM), and United States (US) could provide helpful added context for an international reader, yet Table S6 seems unnecessary.

*We have read through the manuscript and supplied a couple of missing definitions of acronyms, including the first 2 that the referee identified; we consider US to be self-explanatory. Table S6 may be unnecessary, but we have retained in in the Supplement in the event that it may be useful to an international reader in the interpretation of Figures 9 and S5.*

**Anonymous Referee #3**

The authors have conducted a very similar analysis to their previous work and I do not see major differences between this work and the previous work. I would also say this approach has not been well accepted by the ozone community. While background ozone is indeed a large contribution to the health thresholds, I do not believe this paper adds significant new information to the scientific literature.

*We have conducted similar analyses in previous work. What is new in the present paper is the application of our analysis approach to the southwestern US, a particularly interesting region not considered previously. Our paper discusses the importance of this extension, demonstrating the following points (among others):*

*1) As we note in the Introduction and Background Section 1, the southwestern US is a region " of particular interest because it is impacted by large background ozone concentrations (e.g., Zhang et al., 2020). Previous work shows that the background contributions to ODVs in some western US regions exceeds 60 ppb (e.g., Langford et al., 2022), and even has approached 70*

*ppb, making achievement of the 70 ppb NAAQS quite difficult in those regions (Cooper et al., 2015)." Sections 4.1, 4.3 and 4.4 further demonstrate and quantify this regional characteristic.*

*2) As we discuss in Section 5.2 and illustrate in Figure 9, the highest observed US ozone concentrations are now primarily confined to the southwestern US including California and Texas.*

*3) As we discuss in Section 5.4 background ozone overwhelmingly dominates during episodes of even the largest observed ozone concentrations in the southwestern US (which has indeed precluded NAAQS achievement; see discussion of referee's point 4. below). Despite this dominance, the US EPA recently downgraded the Denver urban area from a "Serious" to "Severe-15" nonattainment area under the 2008 ozone NAAQS, which will require further reductions in local and regional precursor emissions – reductions that will be very expensive, and, as we show, ineffective in NAAQS achievement.*

*In summary, our paper does indeed add "significant new information to the scientific literature."*

*We cannot control the extent to which our approach is accepted by the ozone community. We do continue to test, refine and apply our approach, and to compare our results to those derived by other approaches (e.g., see Discussion in Section 5.3 and Figures 10 and 11). In any event, we believe that neither the acceptance nor lack of acceptance of a newer approach by an entrenched scientific community can be used as logical argumentation in the debate of an open scientific question.*

The authors have presented a simplistic model that has a number of problematic assumptions.   Mainly:

*Below the referee raises several issues. Most of these issues were raised during reviews of our previous publications. As part of the "test, refine and apply" process that we mention above, these issues have previously been carefully considered and thoroughly addressed. We believe that these comments are representative of a common outlook in our field that observational analysis is too simple to yield useful insights given the complexity of ozone formation and loss in the troposphere and that computational approaches such as CTMs must be used solely. In anticipation of such comments we include discussion in the paper's Introduction and Background Section 1 of how observational analysis and CTMs are complementary rather than one excluding the other. We also included additional, more detailed discussion of the issues raised by the Referee in the Supplement. Below we give responses to each of the referee's comments, with references to that previous work where possible.*

1.  Their model assumes that high ozone days from stratospheric intrusions or other background sources are the same days as high ozone days due to local photochemical production. This is clearly not the case.

*Our model definitively does NOT make such an assumption. Such a comment has been made previously; thus, we attempted to preemptively address this issue by including a detailed discussion of our model assumptions related to this issue in "Section S2. Relationship of US background ODV to ozone exceedance days" of the Supplement; we also include a brief discussion of this issue in the paper's Introduction and Background Section 1: "Furthermore,*

*the ODVs represent rare events that may occur at any time during the warm season when the combined background and anthropogenic contributions maximize. This maximum may not occur on the days when the background contribution is largest (i.e., the days that determine the US background ODV). Thus, for any given extreme ozone episode the sum of US background ODV and the anthropogenic contribution to that episode may not equal the observed ODV; ...."*

2. The model assumes that anthropogenic emissions (of NOx) are approaching zero. This assumes that our inventories are accurate and that sources such as agricultural emissions are going down at the same rate as other emission sources. I do not believe this is a good assumption.

*Our model only assumes that anthropogenic emissions have decreased, but makes no assumption regarding either their approach to zero, or any other quantitative aspect of emissions. As a result, no possible inaccuracy in emission inventories can affect our model. It is well established and widely accepted that industrial and urban US anthropogenic emissions have indeed decreased over the past decades (e.g., Warneke et al., 2012; Pollack et al., 2013). The assumptions in the model regarding the ozone contribution from local and regional photochemistry are limited to the development of Equations 3 and 4; most important is the assumption that this contribution to ODVs has decreased exponentially in response to those precursor emission reductions, i.e. as quantified by the A\*exp(-t/t) term in those equations. Sections S3 through S5 of the Supplement give detailed discussion of the choice and application of this assumed exponential functional form. Earlier discussion of these issues is included in responses to reviews (acp-2018-1174-AC1.pdf; acp-2018-1174-AC2.pdf; acp-2018-1174-AR2.pdf) of our earlier paper (Parrish and Ennis, 2019).*

*Indeed, one of the strengths of our method (based solely on observed O₃) is its independence from the variety of uncertainties inherent in inventories of precursors, which can often differ by 50-80% (Granier et al., 2011). Of course, other models such as all photochemical grid models that require emission quantification as input are subject to such uncertainties.*

*Importantly, we do not assume that agricultural emissions are decreasing as are industrial and urban US anthropogenic emissions (and we do not believe that the temporal variation of these emissions is well established). They are a relatively minor, highly localized influence that we treat in a simplified manner, as touched upon in our response to the following comment. And, in fact, one of the important findings of our work is the identification of regions where background and/or unregulated local emissions from agricultural soils and wildfires are significantly contributing to the ozone extreme values.*

3. The model treats agricultural emissions and wildfire contributions in a extremely simplistic way (e.g. equation 4).

*Our entire analysis approach is simplistic, yet importantly, observation-based. This is by design, which we consider to be an advantage, as it allows the results to inform our understanding of the extremely complex atmospheric system that determines the highest values of tropospheric ozone concentrations. Again we look to a reference like Granier et al. (2011) mentioned above that claims biomass burning emissions can vary between inventories by 50-80%. Thus, even highly sophisticated chemical transport models can simulate impacts that diverge markedly.*

*As discussed in two paragraphs near the end of the paper's Introduction and Background Section 1 and in Section S1 of the Supplement, we view our approach as one model within the hierarchy of models necessary to provide robust, reliable support for ozone air quality policy development. In keeping with this approach, we do treat the relatively minor agricultural and wildfire contributions in a simple manner.*

4. The authors statements about the policy implications are not correct. States can work with the EPA to exclude high ozone data that is not under their control thru the exceptional events policy.   So for example, on line 30:   "Together, the US background plus wildfire contribution approach or exceed the US NAAQS for ozone of 70 ppb (implemented in 2015) and 75 ppb (implemented in 2008); consequently, in the southwestern US NAAQS achievement has been precluded." This is really not true.   The EPA has a mechanism to exclude background O3.  Its the exceptional events rule.   Its not easy to get these excluded, but it is possible.  Many states have used this to exclude days that have either a strong strat influence at the surface or wildfire influence.

*Our statements regarding the policy implications of our analyses have been carefully worded to ensure that they are indeed correct. The above quote taken from our abstract is correct as written, because it is written in past tense (i.e., "… in the southwestern US NAAQS achievement has been precluded.") We do agree that the exceptional events rule of the EPA provides a mechanism through which it is theoretically possible to exclude days with high background ozone. However, the process is so difficult that (as we note in the discussion in the manuscript) it has only very seldom been utilized successfully by southwestern US states. According to the statistics compiled by David et al. (2021), nation-wide during the 18-year 2000-2017 period, the majority (56%) of the total days that exceeded the ozone standards and were also flagged for an exceptional event in the AQS database were in the 5 southwestern states, CA or TX. Of these only ~3% were submitted to EPA and received concurrence. These days comprised 6 separate exceptional events; thus there has been on average only 1 exceptional event every 3 years successfully removed from nonattainment consideration within the seven state region.*

*Importantly, this is not simply an inconsequential issue of verb tenses; since NAAQS achievement has been precluded, the southwestern US is subject to the imposition of very expensive and burdensome additional precursor emission controls, as have been imposed upon the Denver urban area as discussed in Section 5.4 of our manuscript.*

*There is a further issue that the reviewer should consider. We do demonstrate that the US background plus wildfire contributions approach or exceed the US NAAQS for ozone; is it really reasonable and sensible to set a standard that would require some states to demonstrate multiple exceptional events each year to achieve attainment? The reviewer does admit that it's not easy to exclude days.*

For these reasons I recommend the manuscript be rejected.

*Our responses above fully address the reviewer's comments and show that none raises significant issues with our analyses or discussion. Thus, we urge the editor to accept our manuscript for publication.*

**References not included in paper:**

David, L. M., Ravishankara, A. R., Brey, S. J., Fischer, E. V., Volckens, J. and Kreidenweis, S.: Could the exception become the rule? 'Uncontrollable' air pollution events in the US due to wildland fires. Environmental Research Letters 16, no. 3, 034029, 2021.

Granier, C., Bessagnet, B., Bond, T., D'Angiola, A., van der Gon, H. D., Frost, G. J., Heil, A., et al.: Evolution of anthropogenic and biomass burning emissions of air pollutants at global and regional scales during the 1980–2010 period. Climatic change 109, 163-190 2011.

He, J., C. Harkins, K. O'Dell, M. Li, C. Francoeur, K.C. Aikin, S. Anenberg, B. Baker, S.S. Brown, M.M. Coggon, G.J. Frost, J.B. Gilman, S. Kongdragunta, A. Lamplugh, C. Lyu, Z. Moon, B. Pierce, R.H. Schwantes, C.E. Stockwell, C. Warneke, K. Yang, C.R. Nowlan, G. González Abad, and B. McDonald, COVID-19 perturbation on US air quality and human health impact assessment, *PNAS Nexus*, doi:10.1093/pnasnexus/pgad483, 2024.

Jiang Z, et al., Modeling the impact of COVID-19 on air quality in southern California: implications for future control policies. Atmos Chem Phys. 21:8693–8708, 2021.

Pollack, I. B., Ryerson, T. B., Trainer, M., Neuman, J. A., Roberts, J. M. and Parrish D. D.: Trends in ozone, its precursors, and related secondary oxidation products in Los Angeles, California: A synthesis of measurements from 1960 to 2010. J. Geophys. Res.: Atmos 118 (11):5893–911. doi:10.1002/jgrd.50472, 2013.

Warneke, C., de Gouw, J. A., Holloway, J. S., Peischl, J., Ryerson, T. B., Atlas, E., Blake, D., Trainer, M., and Parrish, D. D.. Multiyear trends in volatile organic compounds in Los Angeles, California: Five decades of decreasing emissions. J. Geophys. Res. 117 (D21):D00V17. doi:10.1029/2012JD017899. 2012.

---

## Author Response (AR2)

**Authors' Response to 2ⁿᵈ set of interactive comments by an Anonymous Referee on "Maximum ozone concentrations in the southwestern US and Texas: Implications of growing predominance of background contribution" by D.D. Parrish, I.C. Faloona, and R.G. Derwent**

**Overview of our Response**

The authors appreciate the efforts by an Anonymous Referee regarding our submitted manuscript. We have accepted the Referee's suggestion to add a brief section 5.5. To avoid duplication and minimize added length, some material originally included in the Introduction has been moved to this new section; these changes are indicated in the "tracked changes" copy of the revised manuscript.

Below the Referee's comment is reproduced in *italic* text, both in its entirety and specific extracted phrases to which we respond individually; our responses, both general and specific, are given in plain text.

**Entire comment of Anonymous Referee #1**

*Across all three referee comments there remain significant questions about how this simplistic, observation-based mathematical model can accurately treat particular sources of ozone (or its precursors) such as stratospheric intrusions, wildfires, and more, as well as assumptions about exceptional events. To address these points of contention, even though the article is already very long it may be useful to add a brief section 5.5 to explicitly describe the limitations of the present approach. This section could clearly (re)iterate to the scientific community what this approach does and does not consider and/or accomplish, and how it could be thought of in a complementary fashion to other related studies that either employ CTMs or take more detailed approaches to specific urban environments included in this manuscript. This could also be a place to discuss openly their "test, refine, and apply" approach that was mentioned in response to Referee #3 so that it is clear to readers how this methodology has evolved overtime. Such a section could conclude with recommendations for the scientific community regarding the key knowledge gaps that future research might pursue.*

**Authors' Response:**

The Referee begins by asserting that "*Across all three referee comments there remain significant questions about how this simplistic, observation-based mathematical model can accurately treat particular sources of ozone (or its precursors) such as stratospheric intrusions, wildfires, and more, as well as assumptions about exceptional events.*" It is important to note that only the third reviewer of the initial review cycle asserted that our analytical model was overly "simplistic" and based on "problematic assumptions" to which we gave thorough responses to clarify this mischaracterization of our approach. The main concerns of reviewers 1 and 2 had to do with overall organization and flow of the manuscript and our advocacy of our method in contrast to the much more common use of CTMs. We believe we very conscientiously addressed the issues raised in all three referee comments. As a result no further, unaddressed issue regarding our model was identified in any of the responses. Therefore we believe that it is incorrect for the

most recent Referee to assert that "*there remain significant questions*", without supporting that assertion. In our opinion, there remains no significant question regarding our analysis, to which we have not given a thorough, unchallenged response.

The Referee describes our model as "*simplistic, observation-based* (and) *mathematical*"; while this description is correct, it fails to capture one of its most important strengths, which is its function as an integral part of a conceptual model of tropospheric ozone that intuitively explains the broad features of how ozone sources, sinks and transport processes all interact to establish the observed local, regional and larger-scale spatial distributions, seasonal cycles and long-term temporal changes of ozone. As emphasized by Derwent et al. (2023), such an intuitive model is an essential component of a required modeling hierarchy (Held, 2005) that complements the comprehensive numerical models that aim to simulate in full detail as much of the atmospheric chemistry and dynamics as possible. We have added Section 5.5, which is organized around a discussion of this required model hierarchy and how our present observation-based model fits within that hierarchy. With regard to specific discussion points requested by the Referee:

- *"... explicitly describe the limitations of the present approach."*

The new Section 5.5 contains material that previously was in the Introduction, which includes a general discussion of the limitations of the present approach. Our Supplement includes Sections S1-S6 comprising 10 pages of discussion of specific limitations of our approach. We refer to the material in the Supplement, but do not believe that it would be useful to attempt to synthesize that material in the added "brief section".

- *"... (re)iterate to the scientific community what this approach does and does not consider and/or accomplish, and how it could be thought of in a complementary fashion to other related studies that either employ CTMs or take more detailed approaches to specific urban environments included in this manuscript."*

The material from the Introduction that is now in Section 5.5 discusses how our approach can be thought of as complementary to CTMs. Since our approach is based on observed ozone concentrations, which as we note are literally the integrated result of all atmospheric processes, by its very nature our approach necessarily considers all relevant processes. Limitations on what the model can accomplish are exhaustively discussed in the Supplement (see previous bullet).

- *"... discuss openly their "test, refine, and apply" approach that was mentioned in response to Referee #3 so that it is clear to readers how this methodology has evolved overtime."*

It is not possible to include such a discussion in a "brief section". This process is described in more than 100 pages published in the Supplements of Parrish et al. (2017; 2022), Parrish and Ennis (2009) and the Supplement to the present paper, as well as in the reviews of our previous and present papers; the reviews of Parrish and Ennis (2019) are publicly available from the *ACP* website, as hopefully will be the review of the present paper, but others are not published online.

- *"... conclude with recommendations for the scientific community regarding the key knowledge gaps that future research might pursue."*

In a previous paper Derwent et al. (2023) we describe a process that we believe is essential for establishing an overall science-into-policy assessment for tropospheric ozone. In Section 5.5 we do not repeat that description; rather we briefly discuss some specific knowledge gaps regarding US surface ozone concentrations that require further investigation through both CTM and observational-based modelling approaches.

---

## Author Response (AR3)

**Authors' Response to comments by ACP editors on "Maximum ozone concentrations in the southwestern US and Texas: Implications of growing predominance of background contribution" by D.D. Parrish, I.C. Faloona, and R.G. Derwent**

The authors appreciate the continuing efforts by the ACP editors regarding our paper. We have accepted their suggestion to revise our manuscript to more clearly emphasize the limitations of our 'simple observation-based model'. This revision is included within **Section 5.5. A required modelling hierarchy**, as indicated in the "tracked changes" copy of the revised manuscript.

Below the Editors' comments are reproduced in *italic* text in their entirety, where we have added numbers in **bold** to identify specific limitations identified by one Editor. We give individual numbered responses to each of those comments, again with the specific extracted phrases in *italic* text with some rewording added by us for clarity. Our rewording (within parentheses) and responses are given in plain text.

**Entire comment of Editors**

*I appreciate your responsiveness to most of the issues raised by reviewers 1 and 2 in the first round. The reviewer of the second version raises some concerns that echo those raised by reviewer 3 in the first round and it is my opinion that these issues should be addressed more explicitly than they currently are. I was struggling with this decision so I consulted with ACP Senior Editor Andreas Hofzumahaus who responded as follows:*

*"In my opinion, the manuscript in its latest version is written in a comprehensible manner, as requested by Referee #1 in the second round. The mathematical approach and the assumptions made are well explained. However, the paper gives the overly optimistic impression that the 'simple observation-based model' is as useful as a detailed CTM. Global CTMs attempt to represent the complex reality in great detail and simulate tropospheric ozone. The results can then be used to derive, for example, ozone metrics for policy decisions in a particular country. In contrast, the current paper deals with an empirical method in which the existing long time series of U.S. ozone design values, ODV (the 3-year average of the annual fourth highest daily maximum 8-hour average (MDA8) ozone mixing ratio) are fitted and parameterized by a simple mathematical formula. The time-dependent formula contains a term for the temporal development of the ozone baseline (from long-range transport and chemical formation from natural emissions in the US), a term for ozone due to production from anthropogenic US emissions, and for ozone due to the long-term increase in wildfire emissions. As I understand the paper, the approach is able to reproduce the observed ODVs and allows conclusions to be drawn about the contribution of anthropogenic US emissions, for example, to ozone exceedances. However, there is no clear mention that **(1)** the approach, unlike CTMs, cannot make predictions about the future development of ODVs. **(2)** Neither can it provide detailed understanding of the chemical or physical processes that contribute to ozone formation. It is a descriptive parameterization of ODVs from the past up to the present in the US. **(3)** The 'observation-based model' presented does not simulate atmospheric ozone, but parameterizes a time series of a regulatory parameter (ODV) defined by US policy for the US. **(4)** The criticism of Anonymous Referee #2 on EGUSPHERE-2023-1231 (which was rejected) that the concept and results are therefore applicable only to the US is still valid."*

*In summary, please revise your manuscript to more clearly emphasize the shortcomings of the simplified model.*

**Authors' individual numbered responses to each of the editor comments**

We agree that our model does have limitations as described by two of the editors' (slightly edited) comments, specifically:

**(2) (**Our model cannot) *provide detailed understanding of the chemical or physical processes that contribute to ozone formation. It is a descriptive parameterization of* (an ozone concentration metric) *from the past up to the present.*

**(3)** *The 'observation-based model' presented does not simulate atmospheric ozone, but parameterizes a time series of* (an ozone concentration metric).

We have included a brief discussion of the issues identified in these comments within an expanded, more general emphasis on our model's limitations in the revised Section 5.5. Clearly, without any representation of the great detail of complex reality included in global CTM simulations of tropospheric ozone, our 'simple observation-based model' cannot perform many of the tasks for which detailed CTMs are utilized.

However, we want to make two important observations here. First, linear trend analysis, an even simpler descriptive parameterization widely utilized in observational-based analysis of ambient ozone concentrations (e.g., Tarasick et al., 2019), suffers from these same limitations, yet this technique is widely accepted without objection. Second, our model does have significant (and evidently under-appreciated) skill at reproducing features of the ambient ozone distribution. We have added a quantitative discussion of this skill to Section S1 of our Supplement.

We believe that two of the Editor's hypothesized limitations of our model are incorrect as specifically discussed below. In this regard, there is one issue that we wish to address before beginning that discussion. The editors' comments end with a summary requesting us to "*revise* (our) *manuscript to more clearly emphasize the shortcomings of the simplified model.*" We believe that the term "*shortcomings*", which can mean **imperfections or flaws that detract from the whole** is inappropriate - no such imperfection or flaw has been identified in our model. The model does perform as designed and accurately gives the information for which it was designed to provide. For "*shortcomings*" we substitute "limitations", which we take to mean **the quality or state of being limited**. Note that some of the following rebuttals are important enough that we have included additional discussion in our revised Supplement.

**(4 including part of 3)** *The 'observation-based model' … parameterizes a time series of a regulatory parameter (ODV) defined by US policy for the US. The criticism of Anonymous Referee #2 on EGUSPHERE-2023-1231 (which was rejected) that the concept and results are therefore applicable only to the US is still valid.*

The model is not limited as described in this comment. First, any long-term measurement record of any ambient ozone concentration metric (not just ODVs) can be fitted and parameterized by the same or similar simple mathematical formulae (just as linear trend analysis can be applied to any such measurement records). This is illustrated in Figure 2 of our manuscript where we fit the same simple mathematical formula to multiple percentiles of MDA8 ozone concentrations in order to quantify the full ozone concentration distribution in CA air basins. We have generally focused primarily (but not exclusively) on ODVs because **a)** they accurately represent the maximum 8-hr mean ozone concentrations observed at measurement sites throughout the US,

and **b)** they are conveniently tabulated by the US EPA. Second, application of the model is possible in any region (not just the US) where long-term measurement records exist. For example, a very similar method has been applied to European measurement records of annual maximum 8-hr (AM8) mean ozone mixing ratios (Derwent and Parrish, 2022). It would be of interest to apply the same or similar approach in many other developed and developing countries of the world where appropriate long-term measurement records have been collected. The criticism of Anonymous Referee #2 on EGUSPHERE-2023-1231 was not valid, either when initially stated or at present.

**(1)** (Our) *approach, unlike CTMs, cannot make predictions about the future development of ODVs.*

Our model certainly can make projections (a term we prefer to "predictions") of the future development of any fitted ozone concentration metric based on simple assumptions regarding the future temporal evolution of the background, US anthropogenic and wildfire contributions to the metric. Importantly, such "predictions" from CTMs require detailed assumptions of the future temporal evolution of all relevant aspects of the complex manifold of chemical and physical processes simulated by the CTM, e.g., future evolution of anthropogenic emissions, changing state of the climate, land use, etc. These assumed temporal evolutions are often simply stasis (i.e., not considered at all during the prediction process) and not discussed. Nevertheless, a vast manifold of either explicit or implicit assumptions are required for such CTM predictions.

In previous papers we have made projections using our model. Parrish et al. (2017) made ODV projections for seven southern CA air basins past 2050 (see their Figure 8) and Parrish and Ennis (2019) projected maximum ODVs in eight northeastern US states until 2025 (dashed curves in their Figure 10), when all ODVs in that region were projected to have decreased to below the 2015 ozone NAAQS of 70 ppb. These projections were made for heuristic purposes, under the simplest possible assumption of persistence of evolution of future ODV contributions, i.e. the parameterized temporal evolution of each contribution over past decades was assumed to continue into the future. Notwithstanding the simplicity of the basis of these projections, we have taken the opportunity presented by responding to this comment to evaluate the fidelity of those projections with the 5 to 8 additional years of ODVs now available since those projections were made, and to compare those projections to some CTM projections. Section S1 of the Supplement discusses the details of this analysis, but two aspects of the results are particularly notable: first, most of the more recent ODVs agree well with the projections, with some significant deviations that lead to important insights into local photochemical environments, and second, within the region of the world that has been the most intensive focus of CTM modeling over decades, i.e., the urban regions of southern CA including Los Angeles, ***the simple projections from our observation-based model are significantly more accurate than CTM projections***.

In summary, we now believe we have fully considered the editors' comments and, where appropriate, revised our manuscript to more clearly emphasize the appropriately identified limitations of our simplified model. We urge the editor to now accept our manuscript for publication.

**References not in original manuscript:**

Derwent, R.G., and D.D. Parrish (2022), Analysis and assessment of the observed long-term changes over three decades in ground-level ozone across north-west Europe from 1989 - 2018, *Atmos. Environ., 286* 119222, https://doi.org/10.1016/j.atmosenv.2022.119222.

Tarasick, D., et al. (2019), Tropospheric ozone from 1877 to 2016, observed levels, trends and uncertainties. *Elem. Sci. Anth., 7*, 39, https://doi.org/10.1525/elementa.376.